# Phospholipids are imported into mitochondria by VDAC, a dimeric beta barrel scramblase

Helene Jahn[1], Ladislav Bartoš[2,3], Grace I. Dearden[1], Jeremy S. Dittman [1], Joost C. M. Holthuis [4], Robert Vácha [2,3] ✉ & Anant K. Menon[1] ✉

Mitochondria are double-membrane-bounded organelles that depend critically on phospholipids supplied by the endoplasmic reticulum. These lipids must cross the outer membrane to support mitochondrial function, but how they do this is unclear. We identify the Voltage Dependent Anion Channel (VDAC), an abundant outer membrane protein, as a scramblase-type lipid transporter that catalyzes lipid entry. On reconstitution into membrane vesicles, dimers of human VDAC1 and VDAC2 catalyze rapid transbilayer translocation of phospholipids by a mechanism that is unrelated to their channel activity. Coarse-grained molecular dynamics simulations of VDAC1 reveal that lipid scrambling occurs at a specific dimer interface where polar residues induce large water defects and bilayer thinning. The rate of phospholipid import into yeast mitochondria is an order of magnitude lower in the absence of VDAC homologs, indicating that VDACs provide the main pathway for lipid entry. Thus, VDAC isoforms, members of a superfamily of beta barrel proteins, moonlight as a class of phospholipid scramblases - distinct from alpha-helical scramblase proteins - that act to import lipids into mitochondria.

The double membrane of mitochondria is composed of phospholipids which are supplied by the endoplasmic reticulum (ER) or assembled in situ from ER-derived phospholipid precursors[1–3]. For example, cardiolipin, the signature lipid of mitochondria, is synthesized at the matrix side of the inner mitochondrial membrane (IMM) from the ER-derived phospholipid phosphatidic acid (PA), and subsequently remodeled to its mature form in the inter-membrane space (IMS) by enzymatic exchange of acyl chains with ER-derived phosphatidylcholine (PC)[1–4]. This poses a considerable lipid trafficking problem, as, after delivery to the cytoplasmic face of the outer mitochondrial membrane (OMM) by non-vesicular mechanisms[5], PA, PC, and other phospholipids must cross the barrier of the OMM before moving on to the IMM (Fig. 1a). Previous work suggested that phospholipids are scrambled across the OMM, i.e., flip-flopped across the OMM by an

ATP-independent mechanism[6–10], but the identity of the OMM scramblase(s) is not known.

The Voltage Dependent Anion Channel (VDAC) is an abundant, multi-functional OMM β-barrel protein with three isoforms (VDAC1-3) in mammals and two (Por1-2) in budding yeast[11–16]. It forms the pore through which ATP and other metabolites cross the OMM and plays a key role in apoptosis[11–16]. The different VDAC isoforms also play non-redundant roles in cells[11,15,17]. Previous molecular dynamics (MD) simulations of mouse VDAC1[18] indicated membrane thinning near an outward-facing glutamate residue (E73), situated midway along the β-barrel. This thinning was observed to promote phospholipid flip-flop[18], with headgroups transiently approaching E73 as the lipids crossed from one side of the bilayer to the other. Using coarse-grained MD (CGMD) simulations, we confirmed that the

[1]Department of Biochemistry, Weill Cornell Medical College, New York, NY 10065, USA. [2]CEITEC - Central European Institute of Technology, Masaryk University, Kamenice 5, 62500 Brno, Czech Republic. [3]National Centre for Biomolecular Research, Faculty of Science, Masaryk University, Kamenice 5, 62500 Brno, Czech Republic. [4]Department of Molecular Cell Biology, University of Osnabrück, Osnabrück 49076, Germany. ✉e-mail: robert.vacha@muni.cz; akm2003@med.cornell.edu

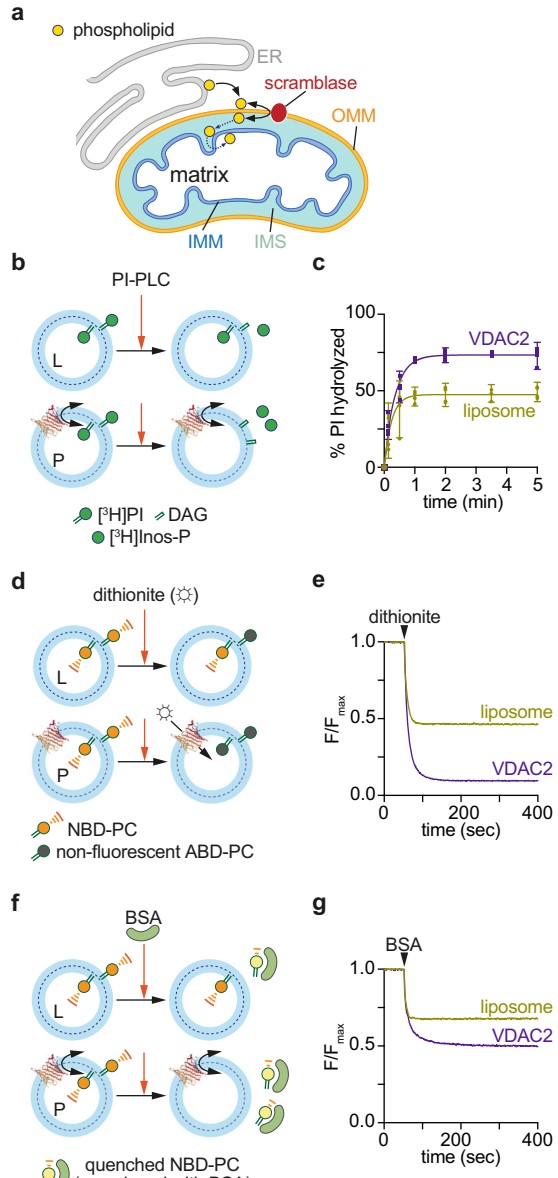

**Fig. 1 | VDAC2 is a phospholipid scramblase. a** Schematic showing phospholipid transport from the ER to all bilayer leaflets of the mitochondrial double membrane. Non-vesicular mechanisms deliver ER-synthesized phospholipids to the cytoplasmic face of the outer mitochondrial membrane (OMM). The lipids are scrambled across the OMM by a lipid transporter (scramblase) before moving through the intermembrane space (IMS) and across the inner mitochondrial membrane (IMM). **b** Scramblase assay using [³H]phosphatidylinositol ([³H]PI). Protein-free-liposomes (L) or VDAC2-proteoliposomes (P) are reconstituted with [³H]PI and probed with PI-specific phospholipase C (PI-PLC), which hydrolyzes [³H]PI to diacylglycerol (DAG) and [³H]inositol cyclic phosphate ([³H]Inos-P). Only [³H]PI molecules in the outer leaflet are hydrolyzed in protein-free-liposomes, whereas all [³H]PI will be hydrolyzed in scramblase-containing vesicles where [³H]PI from the inner leaflet is scrambled to the outer leaflet. **c** Time-course of PI-PLC-mediated hydrolysis of [³H]PI in protein-free-vesicles (liposome) versus vesicles reconstituted with VDAC2 at a theoretical protein/vesicle copy number of 30. Data points are shown as mean and 95% CI, $n = 4$, and fitted to a single exponential function. **d** Channel assay using NBD-PC. Vesicles are reconstituted as in panel b, but with fluorescent NBD-PC instead of [³H]PI. The membrane-impermeant reductant dithionite bleaches approximately 50% of the NBD-PC fluorescence in protein-free liposomes (L), corresponding to those molecules located in the outer leaflet. For vesicles containing a VDAC2 channel, all NBD-PC molecules are expected to be bleached as dithionite can enter the vesicles. **e** Time-course of NBD-PC fluorescence (normalized to the starting value) in liposomes and VDAC2-vesicles on adding dithionite. **f** Scramblase assay using NBD-PC. Vesicles are reconstituted as in panel d and treated with fatty acid-free BSA which extracts NBD-PC from the outer leaflet. In complex with BSA, NBD-PC fluorescence is partly quenched, ~60% lower than when it is in the membrane. For protein-free liposomes, fluorescence is expected to drop by ~30%, whereas for scramblase-containing vesicles fluorescence is expected to drop by ~60%. **g** Time-course of NBD-PC fluorescence (normalized to the starting value) in liposomes and VDAC2-vesicles on adding BSA. Assays similar to those shown in panels e and g were performed 12 times with similar results.

## Results

### VDAC2 scrambles phospholipids in reconstituted vesicles

To test if human VDAC proteins have scramblase activity we used a previously described assay[19] (Fig. 1b). We purified correctly folded human VDAC2 after producing it recombinantly in *E. coli* (Fig. S2) and reconstituted it into large unilamellar vesicles composed mainly of PC with a trace quantity of [³H]inositol-labeled phosphatidylinositol ([³H]PI). Protein-free vesicles were prepared in parallel. On adding bacterial PI-specific phospholipase C (PI-PLC) to the protein-free vesicles we observed rapid ($t_{1/2}$~10 s) hydrolysis of ~50% of the [³H]PI, corresponding to those molecules residing in the outer leaflet of the vesicles and therefore accessible to the enzyme. However, in samples reconstituted with VDAC2, the extent of hydrolysis increased to ~75% (Fig. 1c) indicating flipping of [³H]PI molecules from the inner to the outer leaflet in a large fraction of the vesicles, at a rate ($t_{1/2}$~14 s) comparable to that of the hydrolysis reaction. As flipping was observed in the absence of any metabolic energy supplied to the system, this result indicates that VDAC2 is a scramblase (Fig. 1c).

We further characterized lipid scrambling by VDAC2 using fluorescence-based methods[20–22]. VDAC2 was reconstituted into vesicles containing a trace quantity of 1-myristoyl-2-C6-NBD-PC, a fluorescent PC analog. As VDAC2 is a channel/pore, reconstitution efficiency was assessed using dithionite-mediated bleaching of NBD-PC (Fig. 1d). All NBD-PC molecules are expected to be bleached in a vesicle that contains a VDAC2 channel, which permits dithionite entry, whereas only those molecules in the outer leaflet will be bleached in vesicles lacking a channel. Figure 1e shows that >90% of the NBD-PC is bleached, indicating efficient reconstitution of VDAC2 (in comparison, ~53% bleaching was observed in protein-free vesicles, as expected for 150 nm-diameter vesicles). Next, scramblase activity was assayed using fatty acid-free bovine serum albumin (BSA), which effectively extracts NBD-PC molecules from the outer leaflet of the vesicles, resulting in a decrease in fluorescence as the quantum

membrane-facing surface of VDAC1 provides a low energy path for phospholipid transit across the bilayer, populated by polar residues and centered on E73 (Fig. S1). These observations suggested that VDAC1 - and perhaps other VDAC isoforms - may be able to facilitate rapid phospholipid flip-flop, thereby providing the scramblase activity needed to move lipids across the OMM for mitochondrial function.

Here we show that human VDAC1 and VDAC2, as well as their yeast ortholog Por1, are phospholipid scramblases: on reconstitution into synthetic vesicles these proteins catalyze rapid transbilayer translocation of phospholipids by a mechanism that is unrelated to their channel activity. We also show that VDACs represent the main mechanism by which phospholipids cross the OMM as the absence of VDAC homologs in yeast mitochondria leads to an order-of-magnitude reduction in the transport rate. However, the originally hypothesized transport pathway centered on residue E73 appears not to play an important role. Biochemical studies and CGMD simulations indicate that fast scrambling requires VDAC dimerization, such that phospholipids transit between the leaflets of the bilayer at a specific dimer interface where polar residues induce large water defects and bilayer thinning.

efficiency of NBD-PC complexed to BSA is ~60% lower than that of NBD-PC in the membrane[20–22](Fig. 1f). Thus, quantitative extraction of outer leaflet NBD-PC from protein-free liposomes causes ~30% reduction in fluorescence (Fig. 1g). For vesicles reconstituted with VDAC2, the observed fluorescence reduction is ~55% (Fig. 1g), the extent predicted if ~90% of the vesicles include a scramblase, consistent with the reconstitution efficiency deduced from the channel assay. Curve fitting revealed that fluorescence loss on BSA treatment of VDAC2-containing vesicles is characterized by a rapid phase ($t_{1/2}$~5–10 s, also seen in protein-free samples) corresponding to the extraction of NBD-PC located initially in the outer leaflet, followed by a slower phase ($t_{1/2}$~50 s) which we attribute to trans-bilayer movement of NBD-PC.

To test whether the ability to scramble lipids is unique to the VDAC β-barrel, we purified an unrelated β-barrel protein, Pet464 (Fig. S3a), the 12-stranded β-barrel portion of the *E. coli* Pet autotransporter[23,24]. Vesicles reconstituted with Pet464 had channel activity as expected, albeit with a lower dithionite permeation rate compared with VDAC2 (Fig. S3b), but no scramblase activity (Fig. S3c). Consistent with these data, previous work showed that OmpT, a 10-stranded bacterial β-barrel also lacks scramblase activity[25]. These results indicate that scrambling is not a general property of β-barrel proteins but rather a specific property of VDAC2. We conclude that VDAC2 is a scramblase, capable of transporting both anionic (PI) and zwitterionic (PC) phospholipids.

## VDAC dimers support fast scrambling

To investigate the mechanism of VDAC2-mediated phospholipid scrambling we tested the effect of disrupting the predicted lipid transit pathway (Fig. S1) by substituting the membrane-facing glutamate residue (E84 in human VDAC2) with leucine. We found, unexpectedly, that the E84L mutation had no detectable impact on the ability of VDAC2 to scramble NBD-PC on the timescale of our experiment (Fig. S4), suggesting that the hypothesized pathway is not the principal avenue for lipid scrambling. We therefore considered the alternative possibility that scramblase activity may depend on VDAC's quaternary structure.

VDAC dimers and oligomers have been suggested to be physiologically important, for example in apoptosis, and their formation is regulated by various factors, including pH[26–32]. These oligomeric states have been visualized in the OMM by atomic force microscopy[33,34] and cryoelectron microscopy[35,36], as well as in detergent solution and after reconstitution into vesicles[26,28–31,37,38]. To establish the oligomeric state in which VDAC reconstitutes into vesicles in our protocol, we determined vesicle occupancy statistics[39–42]. We varied the protein/phospholipid ratio (PPR) of the preparation by reconstituting different amounts of VDAC2 into a fixed quantity of lipid vesicles and used the channel assay to determine the fraction of vesicles, at each ratio, that had been functionalized with a VDAC2 channel. The same analyses were also performed using human VDAC1, which we similarly produced in *E. coli* (Fig. S2). As shown in Fig. 2a, b, both VDAC2 and VDAC1 readily functionalize vesicles with channels, with Poisson analysis indicating that they reconstitute as higher-order structures that we estimate to be decamers (Fig. 2c). This could be because the proteins are intrinsically multimeric as purified (see below) and/or multimerize during detergent removal en route to reconstitution[26].

The same vesicle preparations were next taken for scramblase assays. VDAC2 showed robust scramblase activity as expected, the fraction of functionalized vesicles scaling with the PPR as seen for the channel assays (Fig. 2d). However, VDAC1 was surprisingly less potent (Fig. 2e). To evaluate this difference quantitatively, we graphed the fraction of scramblase-active vesicles versus the fraction of channel-active vesicles for both isoforms (Fig. 2f). For VDAC2, these fractions were highly correlated, with ~85% of the channel active vesicles displaying scramblase activity over the range of the PPRs that we tested. However, for VDAC1, scramblase activity was essentially undetectable

on the timescale of the assay until sufficient protein was reconstituted to populate about a third of the vesicles with channels. As the fraction of channel-active vesicles increased beyond ~0.3, the fraction of scramblase-active vesicles also increased but less efficiently (correlation ~70%) than in the case of VDAC2 (Fig. 2f).

To understand the molecular basis for the difference in the behavior of these otherwise highly similar isoforms of VDAC (75% identity, 91% positive match[17,29]), we used chemical crosslinking to probe their quaternary structure after reconstitution. VDAC1- and VDAC2-containing vesicles were treated with EGS, an amine-reactive crosslinker which has been previously used to determine the oligomeric nature of VDAC[30,38](Fig. S5a). Immunoblotting revealed that in the absence of the crosslinker, both proteins were detected mainly as monomers, with a small fraction of VDAC2 forming SDS-resistant dimers. However, the crosslinker efficiently captured dimers and multimers of VDAC2 (Fig. 2g) whereas VDAC1 was recovered mainly in monomeric form, with a small fraction of dimers evident in a higher exposure of the immunoblot (Fig. 2h). We infer that despite their similar incorporation into vesicles as multimers (Fig. 2c), VDAC2 molecules remain associated in the membrane such that complexes can be captured by EGS crosslinking, whereas VDAC1 molecules dissociate, thereby losing their ability to facilitate fast scrambling (assays run over an extended time frame reveal that VDAC1 scrambles lipids slowly, with a long half-time of ~4 h (Fig. S6)). These data suggest that a dimer or higher order multimer of VDAC is necessary for rapid scrambling.

To determine whether dimerization is sufficient for scrambling, we took advantage of the fact that covalent VDAC1 dimers can be formed by crosslinking in detergent. The quaternary structure of VDAC1 is affected by LDAO detergent concentration, with higher order structures predominating at low detergent concentration. These structures dissociate to produce monomers and small oligomers when LDAO concentration is increased to 1% as seen by size exclusion chromatography (Fig. S7a) and EGS crosslinking (Fig. S7b). Building on this information, we identified conditions in which we could crosslink LDAO-solubilized VDAC1 efficiently with EGS (Fig. 3a). Immunoblot analysis revealed that the crosslinked sample, VDAC1[x], consisted principally of dimers, with a small fraction of higher order multimers (Fig. 3a). The EGS-captured dimers likely correspond to VDAC1 protomers oriented in parallel, interacting via strands β1,2,18,19 (Fig. S5c, d), similar to the common dimeric structure reported for human and rat VDAC1[37,38], and zebrafish VDAC2[29]. Importantly, the parallel orientation of the VDAC1 protomers within the dimer structure is supported by spectroscopic analyses of LDAO-solubilized mouse VDAC1[31] and structural studies of human VDAC1 purified in LDAO[37].

Reconstitution of VDAC1[x] and mock-treated VDAC1 showed that they both functionalized vesicles efficiently as evinced by channel activity (Fig. 3b). However, whereas the mock-treated protein was barely active as a scramblase, VDAC1[x] showed robust scrambling activity when reconstituted at the same PPR (Fig. 3c). Quantification of the results (Fig. 3d) showed that ~90% of channel-active vesicles populated with VDAC1[x] are also scramblase active, whereas this number is only ~30% in the case of reconstituted VDAC monomers. We note that VDAC1[x] scrambles lipids ~2-fold more slowly than native dimers/multimers of VDAC2 (Fig. 3c vs 1g), perhaps because of the physical constraints imposed by crosslinking.

To extend our results with EGS, we used DSP, a cleavable crosslinker containing a dithiothreitol (DTT)-susceptible disulfide bond (Fig. S5b). On crosslinking LDAO-solubilized VDAC1 with DSP, we obtained mainly dimers (Fig. S5e), which could be restored to monomers upon DTT treatment (Fig. S5e). DSP-crosslinked VDAC1 had scramblase activity (Fig. S5f, g), like VDAC[x], which was largely lost on DTT-treatment of the reconstituted sample (Fig. S5f, g). We conclude that VDAC dimerization is both necessary and sufficient to generate an efficient scramblase, and that the dimer is the likely minimal functional oligomeric state of the protein.

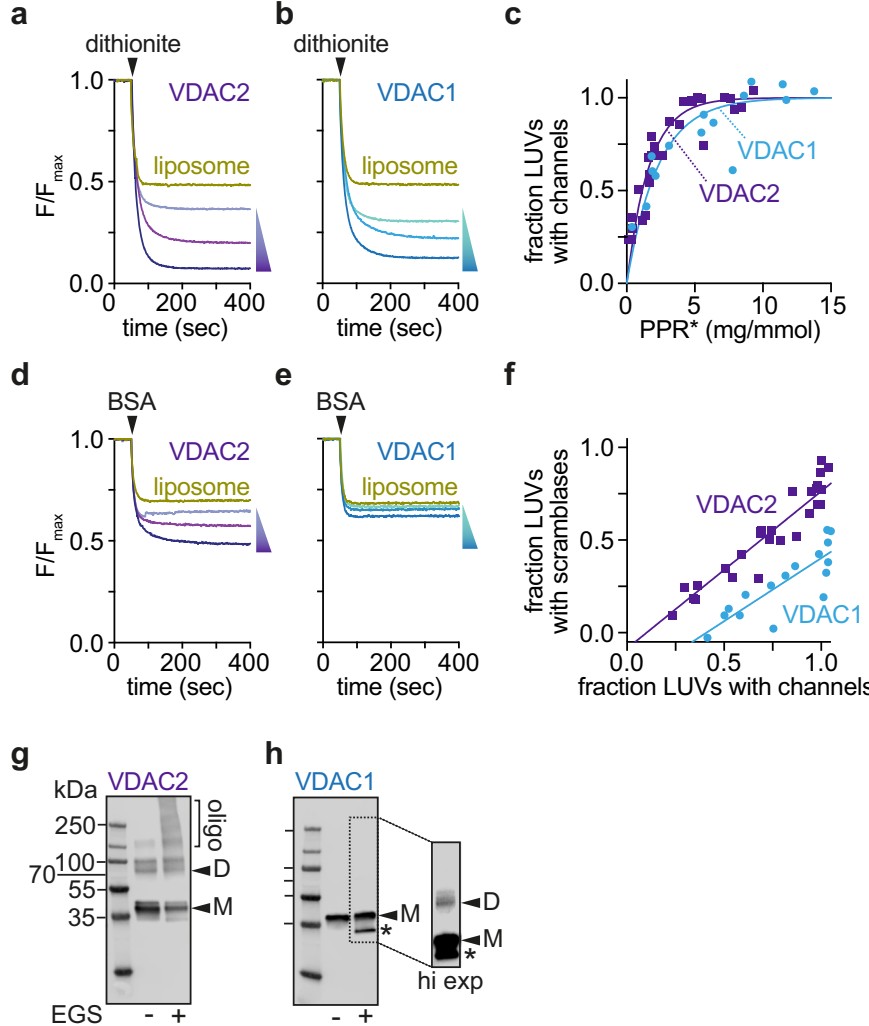

**Fig. 2 | Native VDAC1 scrambles phospholipids poorly.** Representative traces corresponding to channel assays performed on vesicles reconstituted with different amounts of VDAC2 (**a**) or VDAC1 (**b**) (Shown are protein concentrations corresponding to theoretical protein/vesicle copy number of 30, 10 and 2). **c** Protein-dependence plot showing functionalization of vesicles with channel activity, i.e., fraction of large unilamellar vesicles (LUVs) with at least one channel. PPR*, PPR (protein/phospholipid ratio (mg/mmol)), corrected to eliminate the contribution of empty vesicles[41]. The data were analyzed according to a Poisson statistics model for reconstitution of proteins into individual vesicles. A similar mono-exponential fit constant was obtained for both proteins (~2 mg/mmol). Representative traces corresponding to scramblase assays performed on vesicles reconstituted with different amounts of VDAC2 (**d**) or VDAC1 (**e**) as in (**a** and **b**). **f** Correlation between the fraction of LUVs showing channel activity (dithionite assay) and vesicles with scramblase activity (BSA-back extraction assay). Data points correspond to LUVs reconstituted with different amounts of protein ($n = 4$, including two independent reconstitutions). **g**, **h** Crosslinking of VDAC proteins after reconstitution into vesicles. Reconstituted LUVs were treated with EGS to crosslink proteins in proximity. The samples were analyzed by SDS-PAGE immunoblotting using antibodies against the N-terminal His tag. Reconstituted VDAC2 shows a significant population of dimers/multimers (**g**) whereas reconstituted VDAC1 is predominantly monomeric (**h**, side panel shows a brighter signal to enable visualization of a faint dimer band).

## CGMD simulations reveal lipid transit at a specific dimer interface

Our reconstitution experiments with VDAC1 indicate that it functionalizes vesicles efficiently with channel activity but does not facilitate scrambling unless it is dimerized by crosslinking prior to reconstitution or reconstituted at a high PPR to promote dimer formation in situ. Two useful corollaries emerge from this result. First, monomeric VDAC1 provides a potent negative control for our scramblase assay, reinforcing our result with the Pet464 β-barrel (Fig. S3c) - thus, mere reconstitution of a β-barrel is not sufficient for rapid scrambling. Second, the inability of monomeric VDAC1 to promote rapid scrambling indicates that the VDAC pore does not participate directly in this process, and that the pathway for transbilayer lipid transit relies on unique features of VDAC's membrane-facing surface created by dimerization. To understand how VDAC dimers scramble phospholipids we used CGMD to simulate different dimers. We chose two

symmetric dimers, with interfaces mediated by strands β1,2,18,19 (dimer-1) (Fig. 4a, b) as previously reported[29,37,38], and β14-17 (dimer-3), a novel synthetic configuration in which the E73 residue is positioned distal to the interface in each protomer (Fig. S8a). We observed a high scrambling rate for dimer-1 (Fig. 4c, d) over 10 μs of simulation time, with PC molecules moving along the edge of the interface while interacting with both protomers (Fig. 4e, f and Movie S1). In contrast, a VDAC1 monomer supported only slow scrambling, as noted experimentally (Fig. S6), with lipids moving along the originally proposed E73-centered transit path (Fig. S1). Unless restrained, dimer-1 re-oriented during the simulation to dimer-1*, resembling the dimer previously reported by ref. 31, with a symmetric interface mediated by strands β2-4 (Fig. S8b). Upon re-orientation of dimer-1 to dimer-1*, the scrambling rate fell ~4-fold (Fig. 4c, d). Dimer-3 was largely ineffective (Figs. 4d, S8c), scrambling lipids at a rate only marginally higher than that seen for two individual VDAC1 monomers. These results suggest

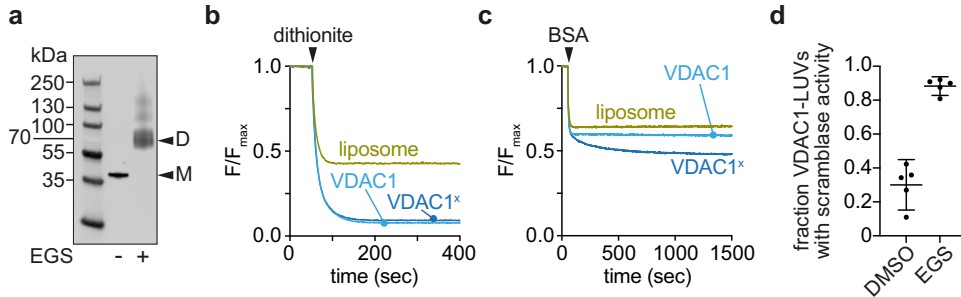

**Fig. 3 | Crosslinked VDAC1 dimers have phospholipid scramblase activity.**
**a** VDAC1 in LDAO was cross-linked using EGS, or mocked treated with DMSO, and visualized by SDS-PAGE immunoblotting using antibodies against the N-terminal His tag. Representative dithionite (**b**) and BSA-back extraction (**c**) traces with cross-linked (VDAC1$^x$) or mock treated (VDAC1) samples reconstituted into LUVs at a theoretical copy number of 30 proteins/vesicle. Normalized fluorescence traces are

shown. These assays were performed 5 times with similar results. **d** Fraction of VDAC1-vesicles with scramblase activity (reconstituted at a theoretical protein/vesicle copy number of 30). Data points shown as individual points, as well as the mean ± 95% CI, $n = 5$, $p < 0.0001$ (two-tailed unpaired $t$ test). After EGS cross-linking, nearly all vesicles that show channel activity also have scramblase activity.

that not all dimer interfaces support rapid scrambling. In support of this conclusion, we experimentally identified a VDAC1 mutant (VDAC1-5V) which, although highly multimeric after reconstitution as revealed by EGS crosslinking (Fig. S9a), scrambled lipids poorly (Fig. S9). We conclude that a specific dimer interface is required to promote rapid scrambling.

The dimer-1 interface possesses numerous polar residues (Fig. 4b), but these are clearly insufficient to promote fast scrambling by themselves because the monomeric protein scrambles lipids slowly (Fig. 4c, d, Fig. S6). We therefore considered that the individual polar faces must synergize at the interface, possibly in concert with the membrane, to create a low-energy trans-bilayer path for lipids. Indeed, we found that the bilayer was much thinner adjacent to the interface, reaching just over 2 nm (Fig. 4g, h), and exhibited a large degree of water penetration (Fig. 4I). These features could also be seen with dimer-1*, but to a smaller extent, correlating with its weaker scrambling activity (Fig. S8f, g). In contrast, the membrane was only slightly perturbed in the vicinity of dimer-3 (Fig. S8f, g) and the monomeric protein (Fig. 4g, i). We hypothesized that scrambling efficiency would be reduced if the six polar residues in the dimer-1 interface (T77, S43, T33, S35, Y247, Q249 (Fig. 4b)) were to be replaced with valine. We simulated this construct (dimer-1-mutant) and observed that its ability to thin the membrane and promote water permeation was intermediate between that of dimer-3 and dimer-1* (Fig. S8f, g), and that it is a poor scramblase with a scrambling rate comparable to a single monomeric VDAC (Fig. S8d, e). Finally, we evaluated the interactions of the lipid with VDAC, membrane, and solvent, during translocation along the dimer-1 interface. Fig. S10 shows that there is a substantial stabilizing interaction between the translocating lipid and VDAC1 along the scrambling pathway, suggesting an important role for lipid-VDAC interactions in decreasing the flip-flop barrier. This role is further supported by a simulation in which the attractive dispersion interactions between the lipid and VDAC were disabled, resulting in an increase in the energy barrier for lipid scrambling (Fig. S10a).

**Mitochondrial lipid import is slowed >10-fold in the absence of VDAC**
Our reconstitution data and in silico analyses clearly show that specific VDAC dimers are efficient scramblases. To quantify the role of VDACs in scrambling phospholipids across the OMM, we turned to the yeast *Saccharomyces cerevisiae*, an organism with VDAC orthologs (Por1 and Por2) that share 70% sequence similarity to human VDAC[14]. We first tested whether yeast VDACs have scramblase activity, and for this we chose to investigate Por1, which is at least 10-fold more abundant than Por2. We over-expressed Twin-Strep-tagged Por1 (Por1-TS) in yeast, and purified it by affinity chromatography, followed by size exclusion

(Fig. S11a). Using the methods outlined in Figs. 1d, f, 2g, we found that Por1-TS had both channel and scramblase activities (Fig. S11b, c), and that the reconstituted sample contained a significant proportion of dimers (Fig. S11d). These data indicate that Por1, like VDAC2, is sufficiently dimeric to not require prior crosslinking for scramblase activity (we note that a small amount of endogenous Por1 co-purified with Por1-TS (Fig. S11a), consistent with its ability to form dimers). Thus, Por1 is a scramblase.

To test the role of VDACs in scrambling phospholipids across the OMM we assayed the ability of isolated yeast mitochondria to convert exogenously supplied phosphatidylserine (PS) to phosphatidylethanolamine (PE), a multi-step process requiring transport of PS across the OMM to the site of PS decarboxylase (Psd1) in the IMS (Fig. 5a)[2]. We added NBD-PS to wild-type mitochondria and observed a time-dependent conversion to NBD-PE as visualized by TLC (Fig. S12a, WT). As expected, no conversion was observed in mitochondria prepared from *psd1Δ* cells (Fig. S12a, *psd1Δ*). The latter result additionally indicates that our preparations are not contaminated with Psd2, a PS decarboxylase localized to the late secretory pathway. We next prepared mitochondria from *por1Δ*, *por2Δ*, and *por1Δpor2Δ* cells, adjusted the preparations to have similar concentrations (confirmed by immunoblotting against the IMM and OMM proteins Psd1 and Tom70, respectively (Fig. S12d)), and measured the rate of conversion of NBD-PS to NBD-PE. Thin layer chromatograms corresponding to the assays are shown in Fig. S12a (lower panels). Quantification of these data yielded time courses (Fig. S12e), and half-times of transport (Fig. S12f). Mitochondria lacking Por1 produced PE at a significantly lower rate than wild-type mitochondria (Fig. S12f), consistent with a role for VDAC in facilitating efficient trans-bilayer movement of PS across the OMM; elimination of Por2, alone or in combination with Por1, did not produce a detectable change in rate consistent with its relatively low abundance. As explained below, the lower rate of PE production in *por1Δpor2Δ* yeast is because the rate of scrambling decreases substantially, such that it becomes lower than the rate of PS decarboxylation.

Because the conversion of NBD-PS to NBD-PE is a multi-step process (Fig. 5a), the rate of NBD-PE production in the assay is only indirectly correlated with the rate at which NBD-PS is transported across the OMM. To tease out how the elimination of Por1/Por2 affects scrambling across the OMM at least two additional contributions to the aggregate kinetics must be considered: deposition of NBD-PS into the OMM outer leaflet and its enzymatic decarboxylation within the IMS by Psd1 (Fig. 5a). As the final decarboxylation step is likely to be rate-limiting, large changes in the rate of transport of NBD-PS across the OMM would be expected to have relatively modest corresponding changes in the rate of NBD-PE production. To quantify the

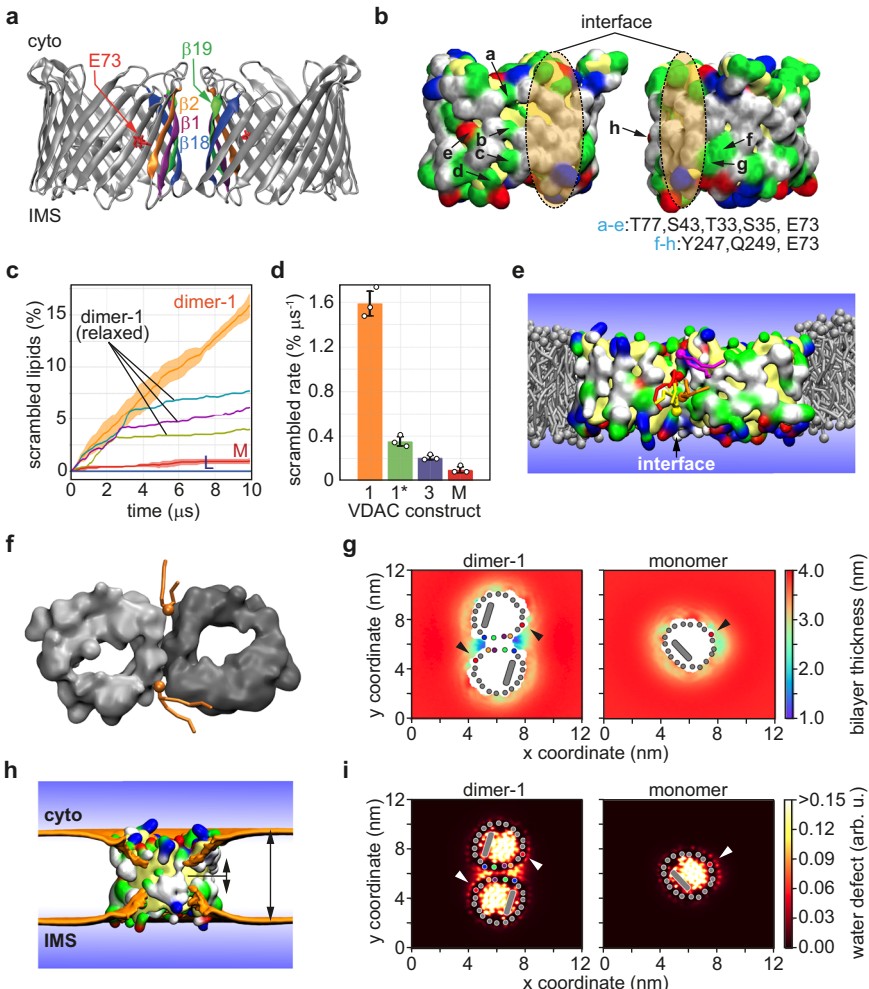

**Fig. 4 | Coarse-grained MD simulations of scrambling by a specific VDAC1 dimer. a** Structure of VDAC1 dimer-1. The β-strands at the interface are indicated, and the E73 residue on the β4-strand is marked. Dimer-1* and dimer-3 are shown in Fig. S8. **b** Surface representation (white, hydrophobic; yellow, backbone; green, hydrophilic; blue, positive; red, negative) showing each protomer, rotated 90° to expose the interface, with the contact region marked with a shaded ellipse; polar residues flanking the interface are indicated. Residues with side-chains oriented towards the pore, are colored yellow. **c** Percentage of lipids scrambled as a function of simulation time. The graphs show the scrambling activity of dimer-1, monomer (M) and protein-free membrane (L) (average over 200 ns time intervals, shading = 68% confidence interval; the inner full line shows the running average of the average scrambling rate measured in 3 independent replicas). Unless constrained during the simulation, dimer-1 reorients to form dimer-1*. Three individual runs of unconstrained dimer-1 are shown (dimer-1 (relaxed)), with the change in scrambling rate occurring between 2 and 4 μs coinciding with reorientation. **d** Bar chart showing scrambling rate for dimer-1, dimer-1*, dimer-3 and monomer (M) (mean ± SD, $n = 3$). **e** Snapshot of dimer-1 from the simulation showing phospholipids (multiple colors) transiting between bilayer leaflets along the interface. Membrane phospholipids surrounding the dimer are gray. **f** Top view snapshot of dimer-1 showing representative phospholipids transiting across the bilayer on both sides of the interface. **g** Bilayer thinning at the dimer-1 interface and near E73 in monomeric VDAC1 (thickness indicated by the color scale at right). Top views are shown, β-strands at the interface are indicated as colored dots (same color scheme as in **a**); the red dot indicates the β4-strand where E73 is located (arrowheads); the N-terminal helix is shown as a gray oblong within the VDAC1 pore. **h** Snapshot side-view of VDAC dimer-1, with one protomer removed, demonstrating membrane thinning at the interface. The ochre surface indicates the average positions of lipid phosphates from both membrane leaflets. The protein is shown in surface representation. **i** Water penetration into the membrane (water defect, color scale shown at right) in the vicinity of the dimer-1 interface and monomeric VDAC1, as indicated.

contribution of VDAC-mediated lipid scrambling to the overall process, we developed a simple kinetic framework comprising PS capture, OMM transit, and enzymatic conversion by Psd1, and fit this model to the measured time course of PE production using either wild-type or *por1Δpor2Δ* mitochondria. A minimal scheme required a four-state model accommodating three distinct transitions (Fig. 5a): first, NBD-PS is reversibly deposited onto the OMM surface with kinetics characterized by rate constants $k_0$ and $k_1$ assuming pseudo-first order kinetics with an excess of available OMM binding capacity (Fig. S12b, c); second, VDAC reversibly transports NBD-PS across the OMM with out→in and in→out rate constants $k_2$ and $k_3$, respectively; third, NBD-PS is irreversibly decarboxylated to generate NBD-PE with an effective rate constant $k_4$. Note that $k_4$ incorporates the intrinsic kinetics of the

Psd1 enzyme, as well as any contributions from NBD-PS adsorption/desorption kinetics within the IMS, into a single effective rate constant.

We determined the kinetic parameters of the four-state model for wild-type mitochondria as follows. Using liposome-based assays (see Materials and Methods) we determined $k_0 \sim 1 \, s^{-1}$ and $k_1 \sim 0.05 \, s^{-1}$ (Fig. S12g–j). Assuming $k_2 = k_3 > 0.01 \, s^{-1}$ based on previous studies[9], and leaving $k_4$ unconstrained, we fit the time course of NBD-PE production to obtain $k_2 = k_3 = 0.027 \pm 0.006 \, s^{-1}$ and $k_4 = 0.0018 \pm 0.00005 \, s^{-1}$ (Fig. 5b, c). As porin deletion would not be expected to alter $k_0$, $k_1$, or $k_4$, we next fit the kinetic data for the *por1Δpor2Δ* mitochondria to obtain the corresponding values of $k_2$ and $k_3$. The best-fitting model produced $k_2 = k_3 = 0.0025 \pm 0.0004 \, s^{-1}$, indicating that the rate of lipid scrambling across the OMM decreases by more than 10-fold in the

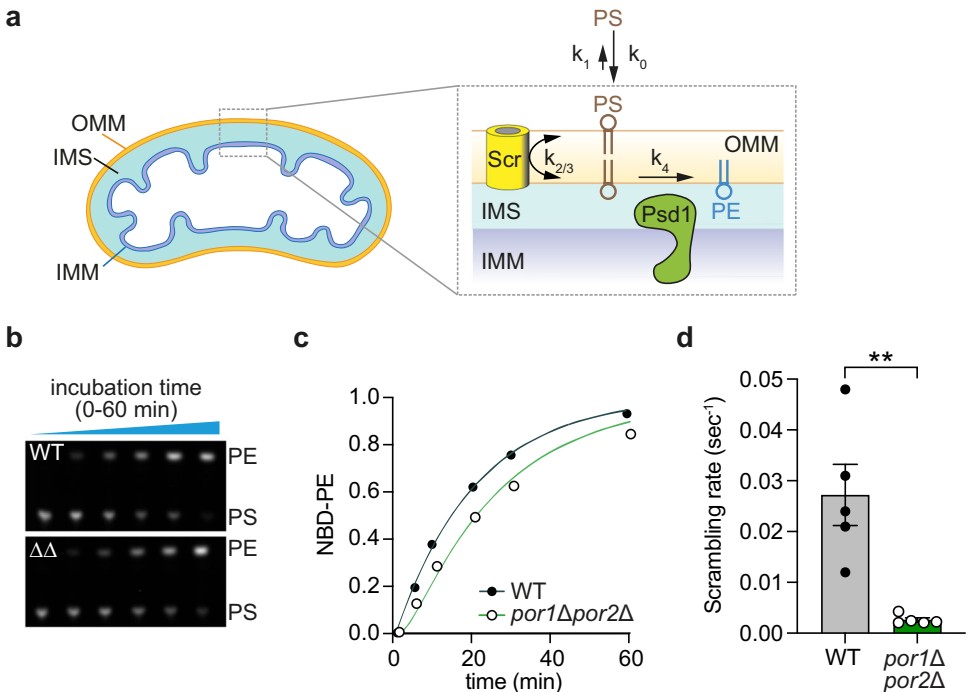

**Fig. 5 | Phospholipid transport across the OMM is slowed more than 10-fold in yeast mitochondria lacking VDAC homologs. a** Assay schematic. NBD-PS (indicated PS) is added to yeast mitochondria. After insertion into the outer leaflet of the OMM, NBD-PS flips reversibly across the membrane where it encounters IMM-localized PS decarboxylase (Psd1) which converts it to NBD-PE (indicated PE). Psd1 can act in trans, hydrolyzing PS in the inner leaflet of the OMM as shown. It can also act *in cis* on NBD-PS molecules that are delivered to the IMS side of the IMM. The decarboxylation of exogenously supplied NBD-PS can be described using a 4-state kinetic model and 5 effective rate constants as shown: deposition of NBD-PS into the OMM ($k_0$), desorption from the OMM ($k_1$), scrambling across the OMM ($k_2$ and $k_3$, presumed to be identical and written as $k_{2/3}$), and irreversible conversion to NBD-PE ($k_4$). **b** Thin layer chromatogram, visualized with a ChemiDoc fluorescence imager, of a decarboxylation assay time-course using mitochondria from wild-type yeast (top panel) and the *por1Δpor2Δ* double mutant (bottom panel, labeled ΔΔ). This assay was performed 5 times with similar results. **c** Time courses of NBD-PS decarboxylation corresponding to (**b**) (the traces are as in Fig. S12, from 2 biological replicates, with at least 2 technical replicates per assay). The data for wild-type mitochondria were analyzed using a 4-state kinetic model (panel a) with $k_0 = 1\,s^{-1}$, $k_1 = 0.05\,s^{-1}$, and $k_2 = k_3 > 0.01\,s^{-1}$ to obtain $k_{2/3}$ and $k_4$. The fitting yielded $k_4 = 0.0018\,s^{-1}$. Data for the *por1Δpor2Δ* double mutant were analyzed using the same kinetic model with $k_0 = 1\,s^{-1}$, $k_1 = 0.05\,s^{-1}$ and $k_4 = 0.0018\,s^{-1}$ to determine $k_{2/3}$. Example time courses of the 4-state model $P_3$ pool using wild-type (black) or porin mutant (green) rate constants are shown with circles representing experimental data (see Methods for details of the model). The data are representative of assays done on two biological replicates with at least 2 technical replicates per assay. **d** Best fitting scrambling rates ($k_{2/3}$) for wild-type and por1Δpor2Δ mitochondria across several experiments obtained as in panel c. The corresponding half-times are 25 and 265 s for wild-type and *por1Δpor2Δ* mitochondria, respectively. **\*\****p* = 0.00738 (2-tailed (unpaired) T test). Error bars correspond to SEM.

absence of VDAC (Fig. 5d). Although our kinetic model is highly simplified, we chose conservative values for the rate constants that likely underestimate the true scrambling rate in wild-type mitochondria. The loss of >90% of the lipid import capability in mitochondria lacking VDACs therefore indicates that VDAC proteins provide the majority of scramblase activity associated with the OMM.

## Discussion

We identified the β-barrel proteins human VDAC1 and VDAC2, as well as their yeast ortholog Por1, members of a large superfamily of beta barrel proteins[12,43], as phospholipid scramblases. Our results indicate, that on specific dimerization, these proteins play a prominent role in transporting anionic and zwitterionic phospholipids across the OMM (Fig. 1a), which is required for mitochondrial membrane biogenesis and mitochondrial function. The unitary rate of VDAC-dimer-mediated scrambling (>$10^3\,s^{-1}$ in reconstituted vesicles) and the abundance of VDACs (>20,000 copies per yeast cell), ensure that VDACs provide the OMM with orders of magnitude more scramblase activity than is needed to supply the mitochondrial network (0.25 fmol phospholipids per yeast cell) with lipids for cell growth and homeostasis[44]. As all phospholipid scramblases identified so far are α-helical membrane proteins, our results introduce the transmembrane β-barrel as a scramblase-active protein fold. The current mechanistic paradigm for α-helical scramblases is that they provide a polar transbilayer pathway which allows phospholipids to traverse the membrane with their headgroup residing within the pathway while their acyl chains extend into the hydrophobic membrane core, as suggested by the credit-card model[45]. This polar pathway can be formed by two or three adjacent membrane helices within a protein monomer as seen for GPCR and TMEM16 scramblases[46,47], between protomers in an oligomer as in bacteriorhodopsin and Atg9[48–50], or at the surface of the protein[47,50,51]. A polar pathway on the surface of VDAC1 and VDAC2, centered on the membrane-facing glutamate, clearly lowers the barrier for lipid scrambling (Fig. S1) but is not sufficient to support fast scrambling (Fig. S6). In contrast, the dimer-1 interface depresses the free energy barrier far more substantially, enabling fast lipid scrambling. The effectiveness of the dimer-1 interface in promoting fast scrambling derives from the synergistic effect of attractive interactions between lipid headgroups and protein residues, and barrier lowering caused by membrane thinning and associated water defects (Fig. S10)[52]. Neither feature alone is sufficient to account for fast scrambling activity. Notably, because of the requirement for quaternary structure in generating a fast scrambling path, it is possible that scrambling at the OMM in cells may be regulated through control of dimerization by other proteins as well as by the lipid environment[26,28,53].

Elimination of both VDAC homologs in yeast slowed scrambling across the OMM by more than an order of magnitude (Fig. 5d), indicating that these proteins are the principal lipid importers of the OMM.

Consistent with this, depletion of VDACs in yeast and mammalian cells caused a reduction in the levels of cardiolipin[54–56], a lipid whose synthesis requires the import of phospholipid precursors from the ER. However, the residual (<10% of total) scramblase activity in *por1Δpor2Δ* mitochondria (Fig. 5d) suggests that there are additional scramblases. Possible candidates are the ubiquitous OMM translocation channel Tom40, a 19-stranded β-barrel which is evolutionarily related to VDAC and shares the same fold[43,57–59]. In support of this idea, Tom40 dimers - like VDAC dimers - have been shown to influence the membrane through bending and destabilization[43,57–59]. Other transmembrane β-barrels of the OMM like Sam50 or Mdm10 might show similar properties[60,61]. The OMM protein MTCH2, recently characterized as a protein insertase involved in the insertion of a variety of α-helical proteins into the OMM[62], has a prominent transmembrane groove containing polar/charged residues, reminiscent of the groove implicated in scrambling lipids according to the credit card model[45]. Our CGMD analyses of monomeric MTCH2 indicate that it scrambles phospholipids at a rate similar to that of dimer-1 (Fig. 4c)[63] (see also ref. 64). Further work will be needed to quantify the relative individual contributions of MTCH2 and the other scramblases mentioned above to the overall scramblase activity in the OMM.

Recent discoveries have advanced the concept that lipid transfer proteins directly engage scramblases[65,66]. Thus, Vps13 and the ERMES complex that are involved in lipid flow between the ER and mitochondria physically interact with the yeast OMM proteins Mcp1 and Mdm10, respectively[66]. Purified Mcp1 was recently shown to have scramblase activity[66] and it remains to be seen whether this is also the case for Mdm10. In agreement with this concept, previous studies placed VDAC at ER-mitochondria contact sites and at hubs of lipid synthesis and distribution[3,54,55]. Thus, in yeast, the VDAC homolog Por1 is thought to interact with the ERMES complex, the Ups-Mdm35 lipid transfer complex as well as Mdm31/32, the latter two being important for phospholipid transport across the IMS[3,54,55]. In this scenario, VDAC constitutes a nexus between the phospholipid transport machineries on both sides of the OMM, scrambling phospholipids to enable their entry(exit) into(from) these machineries.

## Methods
### VDAC purification
VDAC proteins were expressed and purified as described by ref. 67. Chemically competent *E. coli* BL21(DE3)-omp9 cells [F⁻, ompT hsdS_B (r_B⁻ m_B⁻) gal dcm (DE3) ΔlamB ompF::Tn5 ΔompA ΔompC ΔompN::Ω] after ref. 68 were transformed with pCold vectors encoding N-terminal hexahistidine tagged human VDAC1 and VDAC2 isoforms, the point mutant VDAC2 E84L, or the pentamutant VDAC1-5V (T60V, Y62V, E73V, T83V, S101V). All constructs were verified by Sanger sequencing. For VDAC purification, the transformed cells were grown at 37 °C to $OD_{600}$ of 0.6 in 2xYT medium (16 g/l Tryptone, 5 g/l NaCl, 10 g/l Yeast extract) containing 100 μg/ml ampicillin. The cultures were cooled before adding 1 mM IPTG and incubating overnight at 15 °C. Cells were harvested by centrifugation and the cell pellet was resuspended in TEN-buffer (50 mM Tris-HCl pH 8.0, 100 mM NaCl) containing a protease inhibitor cocktail (30 ml of resuspension for 2 L starting cell culture). The cells were disrupted using a probe sonicator and inclusion bodies were collected by centrifugation at 25,000 x g for 30 min at 4 °C. Residual cell membranes were removed by washing the pellet thrice with 15 ml TEN-buffer containing 2.5% v/v Triton X-100, using a Teflon pestle for resuspension followed by centrifugation at 25,000 x g 15 min 4 °C, after which the Triton X-100 was removed by further washing the inclusion bodies thrice in TEN-buffer. Washed inclusion bodies were resuspended in 4 ml TEN buffer for 2 L starting cell culture and 2 ml were used for immediate denaturing by dropwise addition into denaturation buffer (25 mM Na⁺PO₄ pH 7.0, 100 mM NaCl, 6 M guanidine hydrochloride, 1 mM EDTA, 10 mM DTT) to a 10-times dilution. After overnight incubation at 4 °C under constant

stirring, the proteins were refolded by serial 10-times dilutions into first 25 mM Na⁺PO₄ pH 7.0, 100 mM NaCl, 1 mM EDTA, 2.22% LDAO followed by overnight incubation. This material was further 10-times diluted into 25 mM Na⁺PO₄ pH 7.0, 10 mM NaCl, 1 mM EDTA, 0.1% LDAO. After incubation at 4 °C for 4 h or overnight, the solution was filtered, applied onto a cation-exchange column (HiTrap™ SP HP 5 ml (GE Healthcare)), and eluted with a salt gradient (10 mM to 1 M NaCl in the buffer 25 mM Na⁺PO₄ pH 7.0, 1 mM EDTA, 0.1% LDAO, 1 mM DTT). Protein-containing fractions were pooled, concentrated to 500 μl using Amicon Ultra-4 (Millipore) centrifugal filters, and loaded onto a Superdex 200 10/300 25 ml size-exclusion column using SEC buffer (10 mM Tris-HCl pH 8.0, 100 mM NaCl, 0.05% LDAO). Peak fractions were pooled, quantified by absorbance (VDAC1 $\varepsilon_{280}$ = 38,515 M⁻¹.cm⁻¹, VDAC2 $\varepsilon_{280}$ = 37,400 M⁻¹.cm⁻¹) and BCA colorimetric assay, assessed for purity by Coomassie-stained SDS-PAGE and analyzed by Circular Dichroism spectroscopy using Aviv 410 CD instrument. For storage at −80 °C, 10% glycerol was added, and the protein was snap frozen. The typical protein concentration obtained was 1 mg/ml from 1 L of starting culture.

### Yeast VDAC (Por1) purification
*S. cerevisiae* W303 cells were transformed with a pBEVY-GL vector[69] for galactose-inducible expression of C-terminal Twin-Strep-tagged yeast VDAC (Por1-TS), with a TEV protease cleavage site in the linker region between the Por1 sequence and the Twin-Strep tag. The expression construct was a kind gift from Dr. Susan Buchanan (National Institute of Diabetes and Digestive and Kidney Diseases). An overnight culture of the cells in synthetic complete medium minus leucine (SC-Leu) with 2% (w/v) glucose was used to inoculate 500 ml SC-Leu with 3% (w/v) glycerol and 0.1% (w/v) glucose. After overnight growth, cells were pelleted and washed twice in a small volume of yeast extract-peptone (YP) medium (1% w/v yeast extract, 2% w/v peptone) with 2% (w/v) galactose to remove all remaining glucose from the growth medium. Resuspended cells were transferred to 1.5 L of YP galactose (starting $OD_{600}$ ~0.6) and grown overnight at 25 °C (20 h).

Cells were harvested, resuspended to a final volume of 50 ml in 50 mM HEPES pH 7.5, 150 mM NaCl, 1 mM EDTA, 0.1 mM PMSF, Roche EDTA-free Mini cOmplete PIC tablets, and lysed with 5 passes through an Emulsiflex-C3 homogenizer at 25,000 psi. Debris and unbroken cells were removed by centrifuging for 10 min at 3000 x g, and the supernatant was subsequently centrifuged at 185,000 x g for 1.5 h in a Beckman 45 Ti rotor to pellet membranes. The pellet was resuspended in 50 ml 50 mM HEPES pH 7.5, 150 mM NaCl, 1 mM EDTA and homogenized 10 ml at a time in a Dounce homogenizer. LDAO (1% (w/v)) was added, and the sample was stirred for 1.5 h at 4 °C. Insoluble material was pelleted at 110,000 x g for 30 min in a 45 Ti rotor. The supernatant was vacuum filtered through a 0.22 μm membrane and subjected to Twin-Strep-tag/Strep-Tactin XT affinity purification as follows. A 1 ml Strep-Tactin XT 4Flow high-capacity cartridge (IBA Lifesciences 2-5027-001) was washed with 14 column volumes of wash buffer (50 mM Hepes pH 7.6, 150 mM NaCl, 1 mM EDTA, 0.1% LDAO) on an ÄKTA pure™ chromatography system. The 50 ml filtered sample was applied to the resin at 1 ml/min, and then the resin was washed with wash buffer until the absorbance at 280 nm had stabilized. Flow through and wash buffer were collected for SDS-PAGE analysis. Bound Por1-TS was eluted with the addition of 50 mM Biotin in wash buffer (pH 7.6) at 1 ml/min, collecting 1 ml fractions, until absorbance at 280 nm reached baseline. Fractions were analyzed by Coomassie-stained SDS-PAGE. Peak fractions were pooled and concentrated to 500 μl using Amicon Ultra-15 10K MWCO centrifugal filters (Merck Millipore UFC901008; final LDAO = 2%) and further purified by size-exclusion chromatography as described for human VDAC constructs but with 20 mM HEPES pH 7.6, 150 mM NaCl, 0.05% LDAO as SEC buffer. Por1-TS-containing SEC fractions were concentrated to 200 ng/μl (quantified by comparison with BSA standards on Coomassie-stained SDS-PAGE)

and snap-frozen in 10% (w/v) glycerol for storage at −80 °C. Mass spectrometric analyses of the purified sample were performed at the Weill Cornell Medicine Proteomics and Metabolomics Core Facility.

## Pet464 purification

The purified Pet464 protein in 50 mM Tris, pH 8.0, 150 mM NaCl, 0.05% LDAO, and the corresponding plasmid pETT22bbPet464ββ containing the Pet protein with a passenger domain truncation to 464 amino acids was a kind gift from Matthew Johnson and Denisse Leyton of the Australian National University[24]. The protein was prepared as described[24]. Formation of inclusion bodies was induced in BL21(DE3) cells using 0.5 mM IPTG for 4 h. Cells were resuspended in 50 mM Tris-HCl pH 8.0, 150 mM NaCl, 1% Triton X-100 and pretreated with 0.7 mg/ml lysozyme chloride on ice for 10 min, and another 30 min after the addition of DNase1 and 5 mM MgCl. After cell lysis by tip sonication, inclusion bodies were collected by centrifugation at 10,000 x g, 10 min and washed three times in 50 mM Tris-HCl pH 8.0, 150 mM NaCl, 1% Triton X-100 with a final wash excluding Triton X-100. The inclusion bodies were solubilized in 50 mM Tris-HCl pH 8.0, 8 M urea for 2 h on a rotation wheel and subsequentially centrifuged at 30,000 x g, 30 min to remove aggregates. The unfolded protein was snap frozen, and stored at −80 °C. The protein was refolded by a rapid 10-fold dilution into pre-warmed refolding buffer (50 mM Tris-HCl pH 8.0, 150 mM NaCl, 0.5% LDAO) at 35 °C for 10 min under constant shaking. Finally, the β-barrel was purified using first Qiagen Ni-NTA Resin Beads and further size exclusion chromatography in SEC-buffer (50 mM Tris-HCl pH 8.0, 150 mM NaCl, 0.05% LDAO). Fractions were collected, quantified by a wavelength scan ($\varepsilon 280 = 48{,}360\,M^{-1}\,cm^{-1}$), SDS-PAGE, and a BCA protein determination before further use for protein reconstitution.

## Large unilamellar vesicles (LUVs)

POPC and POPG (1435 µl and 160 µl of 25 mg/ml stock solutions in chloroform), together with (1-myristoyl-2-$C_6$-NBD-PC) (195 µl of a 1 mg/ml stock solution in chloroform) as indicated, were added to a round-bottom flask. The solvent was evaporated using a rotary evaporator and the flask was placed overnight in a desiccator attached to house vacuum. The dried lipid film was resuspended by swirling in 10 ml reconstitution buffer (10 mM MOPS/Tris pH 7.0, 100 mM KCl) or crosslinking buffer (10 mM MOPS/KOH pH 7.0, 100 mM KCl) as indicated (lipid concentration 5.25 mM), and incubated at 37 °C for 20 min. The resuspended lipids were briefly sonicated in a bath sonicator, before being sequentially extruded through 0.4 µm and 0.2 µm membranes in a high-pressure lipid extruder (Northern Lipids). The size (150 nm) and polydispersity index (12.5) of the preparation were determined by Dynamic Light Scattering (DLS) with the Litesizer™ 500 instrument. The liposomes had a typical concentration of 3 mM.

## VDAC reconstitution into LUVs

VDAC was reconstituted into pre-formed LUVs by a modification of the method of Brunner and Schenck[70]. Briefly, 800 µl LUVs were destabilized by adding 16 µl 10% w/v Triton X-100 and incubating for 20 min with agitation. VDAC samples were supplemented to 1.05% LDAO and agitated (600 rpm) on a thermomixer at room temperature or, in some instances at 37 °C for VDAC1. The desired concentration of protein in a maximal volume of 100 µl was added to the destabilized vesicles, and the volume made up to 1 ml with SEC buffer or crosslinking-buffer (10 mM MOPS/KOH pH 7.0, 100 mM KCl) containing 0.05% LDAO as indicated, with additional LDAO to ensure equal detergent concentrations in all samples. Samples were incubated for 1 h with end-over-end mixing at room temperature (VDAC2) or in some instances agitated (600 rpm) at 37 °C in the case of VDAC1. Washed Bio-Beads (140 mg) were added, and the samples were agitated (600 rpm) for 20 min at 37 °C in the case of VDAC1, then transferred to the cold room for overnight incubation at 4 °C with end-over-end mixing.

Reconstituted vesicles were separated from Bio-Beads and used immediately for further assays. Protein concentration was determined by SDS-PAGE (Coomassie or Fluorescence Protein gel stain (Lamda Biotech)) in comparison with standards. Lipid concentration was determined by colorimetric assay of lipid phosphorus after the protocol of ref. [71]. The PRR (mg/mmol) of the samples was calculated using experimentally determined values. PRR*[41,48] describing the protein per phospholipid ratio normalized against the fraction of vesicles that cannot be populated by proteins was calculated as

$$PPR^* = \frac{PPR}{(1-R)} \text{ with } R = 2*F_{Dith}^{Min} \tag{1}$$

## Fluorescence assays for channel and scramblase activity

NBD-PC-containing proteoliposomes were assayed for VDAC channel activity with the dithionite assay after the protocol of ref. [22] and scramblase activity with the BSA-back extraction assay after the protocol of ref. [20]. Proteoliposomes were diluted 40-times into HEPES buffer (50 mM HEPES pH 7.4, 100 mM NaCl) in a fluorimeter cuvette, and fluorescence was monitored under constant stirring (900 rpm) at 20 °C in a temperature-controlled Spectrofluorometer FluoroMax + instrument using $l_{ex} = 470$ nm, $l_{em} = 530$ nm, slit widths 2.5 nm. The sample was equilibrated for 5–10 min before proceeding with the assays. For the dithionite assay, 40 µl of 1 M sodium dithionite, freshly prepared in unbuffered 0.5 M Tris, was added. For the BSA-back extraction assay, 40 µl 75 mg/ml fatty acid-free BSA (Calbiochem) in HEPES buffer was added. Collected fluorescence traces were normalized to the initial value $F_{max}$. The fraction (f) of vesicles containing either channel or scramblase activity was calculated as

$$f = \frac{F^L - F^P}{F^L - F^{Min}} \tag{2}$$

with the normalized fluorescence (F) at 350 s after dithionite ($f_{Pore}$) or BSA ($f_{Scr}$) addition for protein-free ($F^L$) or protein-containing liposomes ($F^P$) and $F^{min}$ as the lowest fluorescence signal detectable if all vesicles possess activity while accounting for refractory liposomes. For the dithionite assay, $F_{Dith}^{Min}$ was experimentally determined using the average fluorescence value of proteoliposomes reconstituted at a protein:phospholipid ratio 1:6000, using 15 µg/ml of protein to obtain a theoretical copy number of 30 proteins per vesicle. If the dithionite assay captures all liposomes containing VDAC proteins, the fluorescence signal describing $F_{BSA}^{Min}$ for all proteoliposomes theoretically capable of scramblase activity can be calculated as:

$$F_{BSA}^{Min} = \left(1 - F_{Dit}^{Min}\right)*\phi + F_{Dit}^{Min} \tag{3}$$

where φ is the fluorescence of NBD-PC when it is complexed with BSA compared with its value in the membrane. The value of φ was experimentally determined to be 0.4 using the fluorescence signal after 350 s for protein-free liposomes and comparing dithionite and BSA-back extraction traces with rearrangement of the above equation as follows:

$$\phi = \frac{F_{BSA}^{L} - F_{Dit}^{L}}{\left(1 - F_{Dit}^{L}\right)} = 0.4 \tag{4}$$

The fraction of VDAC proteoliposomes that possess scramblase activity (Q) was determined as

$$Q = \frac{f_{Scr}}{f_{Pore}} \tag{5}$$

Additionally, fluorescence traces were analyzed using one-phase or two-phase exponential decay functions as determined using F-test with GraphPad Prism.

To determine oligomeric species that are reconstituted per vesicle, $f_{Pore}$ was plotted against PPR*, and analyzed according to the Poisson model[22,40,41] by fitting to a one-phase exponential function. The fit constant τ (in units of μg protein/μmol lipids) corresponds to the reconstitution condition where each vesicle has a single functional unit on average. Thus, the protein copy (C) number per 1 μmol lipids was determined as

$$C = τ * \frac{1}{MW} * N_A \qquad (6)$$

with MW = 31 kDa, and the number of vesicles (V) in 1 μmol was determined as

$$V = \frac{N_A * 10^{-6}}{L} \qquad (7)$$

with lipids per vesicle (L) as

$$L = \frac{\left(4π*\left(\frac{d}{2}\right)^2 + 4π*\left(\frac{d}{2} - h\right)^2\right)}{a} \qquad (8)$$

with diameter (d) as 150 nm, bilayer thickness (h) as 5 nm and phospholipid headgroup area (a) as 0.71 nm²[72]. Proteins incorporated per vesicles are thus, $\frac{C}{V}$.

## Scramblase assay using PI-PLC

VDAC proteoliposomes were prepared as described above except that 20,000 cpm per ml of [³H]inositol-labeled phosphatidylinositol ([³H] PI) (American Radiolabeled Chemicals) was included during the reconstitution step. [³H]PI was dried under nitrogen stream and taken up in 50 μl per 1 ml sample preparation in reconstitution buffer (10 mM MOPS/Tris pH 7.0, 100 mM KCl) containing 0.2% w/v Triton X-100. No NBD-PC was present in the samples. Scramblase activity was assayed using PI-specific phospholipase C (PI-PLC purchased from Sigma) as described by ref. 19. To 100 μl aliquots of proteoliposomes 10 μl HEPES buffer (50 mM HEPES pH 7.4, 100 mM NaCl) (or 10 μl 10% w/v Triton X-100 for sample disruption) was added. Finally, 3 μl PI-PLC working solution (10-times dilution into HEPES buffer) was added and the samples were incubated for the indicated time at 25 °C. The reactions were stopped by the addition of ice-cold trichloroacetic acid to a final concentration of 12% (w/v) and placement on ice. Cytochrome c (Sigma) was added to a final concentration of 150 μg/ml for better visualization of the protein pellet. Samples were incubated on ice for 30 min before being microfuged to pellet precipitated material (including non-hydrolyzed [³H]PI). The supernatant containing released [³H]inositol-cyclic phosphate was taken for liquid scintillation counting. Control samples included proteoliposomes disrupted with Triton X-100, proteoliposomes not treated with PI-PLC and protein-free liposomes. Scintillation counts were offset-corrected using the value obtained from non-treated samples, and data were subsequently normalized to the maximum extent of hydrolysis determined from Triton X-100 disrupted samples. The percentage of hydrolyzed PI was graphed as a function of time and traces were fitted to a one-phase exponential association using GraphPad Prism software.

## EGS crosslinking

Ethylene glycol bis(succinimidyl succinate) (EGS; ThermoFisher) crosslinking was done according to the protocol of refs. 27,30 with modifications. VDAC was crosslinked in LDAO or after reconstitution into LUVs prepared in crosslinking buffer (10 mM MOPS/KOH pH 7.0, 100 mM KCl). Samples to be crosslinked in LDAO were applied to a

desalting spin column for buffer exchange to crosslinking buffer containing 0.05% LDAO (additional LDAO was added as indicated). EGS was dissolved in DMSO, diluted 20-times in crosslinking buffer (with additional LDAO as indicated), before being diluted 10-times into VDAC samples (15 μg/ml or 150 μg/ml when used for further reconstitution) to achieve a final 16x mole excess over reactive sites (lysine residues + N-terminus), with final DMSO = 0.5%. Non-crosslinked controls were treated with an equivalent amount of DMSO. After agitation (1000 rpm) in a thermomixer at 20 °C for 40 min, the reaction was stopped by adding 5xLaemmli buffer to 1x (60 mM Tris-HCl pH 6.8, 2% w/v SDS, 10% v/v glycerol, 5% v/v β-mercaptoethanol, 0.01% bromophenol blue) and heating to 95 °C for 3 min. Samples containing human VDAC were taken for immunoblotting using anti-6-His tag Monoclonal antibody (MA1-21315, ThermoFisher) at 1000-times dilution in PBST (137 mM NaCl, 2.7 mM KCl, 10 mM Na₂HPO₄, 1.8 mM KH₂PO₄, 0.05% w/v Tween20, pH 7.3 with NaOH) with 3% BSA followed by the secondary antibody anti-mouse IgG, HRP-conjugate (W4021, Promega) at 10,000-times dilution in PBST. Samples containing yeast porin were taken for immunoblotting using anti-VDAC1/Porin monoclonal antibody (ab110326, abcam) at 1000-times dilution in TBST (50 mM Tris, 150 mM NaCl, 0.1% w/v Tween20, pH 7.6 with HCl) with 5% milk powder, followed by the secondary antibody anti-Mouse IgG, HRP-conjugate, as above at 5000-times dilution in TBST containing 5% milk. Crosslinking and control reactions intended for scramblase activity assays were stopped with 1 M Tris pH 7 to a final concentration of 10 mM, instead of Laemmli buffer, and used for reconstitution into LUVs whilst ensuring the same DMSO and EGS concentrations in control protein-free liposomes. The same protocol was followed for crosslinking with dithiobis(succinimidyl propionate) (DSP; Thermo-Fisher) and samples were taken up in 4xSDS-PAGE buffer to 1x (75 mM Tris-HCl pH 6.8, 2% w/v SDS, 10% glycerol, 0.05% bromophenol blue) instead of Laemmli buffer for immunoblotting. For cleavage of DSP, 50 mM DTT (from 1 M stock prepared in reconstitution buffer) was added and the samples were incubated for 30 min at 37 °C while agitated (600 rpm). Control and non-cleaved samples were treated equally omitting DTT.

## Assay of slow scrambling

To assay slow scrambling, proteoliposomes were prepared without NBD-PC and were subsequently asymmetrically labeled by adding NBD-PC (from an EtOH stock solution) to a final concentration of 0.25 mol% lipid, and final EtOH concentration of 0.4%. After 30 min incubation on ice, the samples were shifted to 20 °C and 50 μl aliquots were taken for BSA-back extraction assay as described above every 2 h for up to 30 h. Normalized fluorescence signal 100 s after BSA addition was used to determine the fraction of NBD-analogs at the outer leaflet setting time point 0 as 100% of analogs present at the outer leaflet. Traces were fitted to a one-phase exponential decay function using GraphPad Prism software.

## Isolation of yeast mitochondria

Crude mitochondria were prepared from yeast after the protocol of ref. 73 with some modifications. Briefly, yeast cells were pre-grown in YPD (1% w/v yeast extract, 2% w/v peptone, 2% w/v dextrose) over a weekend, 10-times diluted into rich-lactate (RL) media (1% w/v yeast extract, 2% w/v peptone, 0.05% w/v dextrose, 2% v/v lactic acid, 3.4 mM CaCl₂, 8.5 mM NaCl, 2.95 mM MgCl₂, 7.35 mM KH₂PO₄, 18.7 mM NH₄Cl, pH 5.5 adjusted with KOH) containing 2 mM ethanolamine and grown for 4 h at 30 °C. 6 OD₆₀₀ units of cells were added to 500 ml RL containing 2 mM ethanolamine and further grown overnight to an OD₆₀₀ of 4 to 5. Cells were harvested, washed with 1 mM EDTA, resuspended in T-buffer (0.1 M Tris-HCl, 10 mM DTT, pH 9.4) at 0.5 g cell pellet/ml, incubated for 10 min at 30 °C, and then harvested and washed in 30 °C warm S-buffer (1.2 M sorbitol, 20 mM KH₂PO₄, pH 7.4 with KOH) at 0.15 g cell pellet/ml. Cells were collected and resuspended in S-buffer

containing 1 mg Zymolyase per g yeast cell pellet at 0.3 g cell pellet/ml and incubated at 30 °C for 1 h. The resulting spheroplasts were diluted in an equal volume of ice-cold S-buffer, harvested at 4 °C, resuspended in ice-cold D⁻-buffer (10 mM MES, 0.6 M sorbitol, pH 6 with NaOH) containing 0.2% w/v fatty acid free BSA and 1 mM PMSF at 0.75 g cell pellet/ml, and disrupted with 10 strokes of a Dounce homogenizer. Mitochondria were collected by differential centrifugation, first, intact cells/spheroplasts were removed via two low speed centrifugation steps at 1400 x g 4 °C for 5 min. Crude mitochondria were pelleted by centrifugation of the resulting supernatant at 10,000 x g 12 min 4 °C. Mitochondria were resuspended in D⁻-buffer, snap frozen and stored at −80 °C. The protein concentration of the preparations was determined using the BCA protein assay in the presence of 0.8% SDS, as well as by SDS-PAGE/Coomassie staining in comparison with BSA standards.

## Transport-coupled PS decarboxylation by yeast mitochondria

Conversion of NBD-PS to NBD-PE by crude mitochondria was assayed according to the protocol of ref. [74] with modifications. Crude mitochondria were diluted to 5 mg/ml in D⁻-buffer. The concentration of mitochondria in different samples was matched by assessing concentrations via immunoblotting with anti-Psd1β 4077[75] in a 1000-times dilution in PBST (137 mM NaCl, 2.7 mM KCl, 10 mM $Na_2HPO_4$, 1.8 mM $KH_2PO_4$, 0.05% w/v Tween20, pH 7.3 with NaOH) containing 3% BSA or anti-Tom70[76] at 20,000-times dilution in PBST with 3% BSA. Both antibodies were a kind gift from S. Claypool (Johns Hopkins University); anti-rabbit IgG, HRP conjugate (W4011, Promega) at a 10,000-times dilution in PBST was used as the secondary antibody. The assay was started by 5-times dilution of the mitochondria into D⁺-buffer (10 mM MES, 0.6 M sorbitol, pH 6 with NaOH, 12.5 mM EDTA) containing 1.25 μM 16:0 – 6:0 NBD-PS (Avanti) and 0.3 μM C6-NBD-PC. Samples were incubated at 30 °C and 100 μl aliquots were taken at the respective time points. To stop the reaction, 750 μl $CHCl_3$:MeOH (1:2, v/v) was added. Subsequentially, 250 μl $CHCl_3$ and 250 μl 0.2 M KCl were added, and the samples were centrifuged at 2200 x g for 5 min at RT. The $CHCl_3$ phase was collected and dried under nitrogen. Samples were taken up in MeOH and spotted onto an activated thin layer chromatography (TLC) silica gel 60 plate (Merck). The plate was developed in $CHCl_3$:MeOH:Acetic Acid:$H_2O$ (25:15:4:2, by volume) and NBD fluorescence was visualized using the BioRad ChemiDoc System. Densitometry of the fluorescence (F) signal was used to determine PS to PE conversion according to

$$PE = \frac{F_{PE}}{F_{PE} + F_{PS}} \tag{9}$$

## Kinetic analysis of transport-coupled PS decarboxylation

The kinetic model is described in Fig. 5a - the model specifies 4 pools of lipid, and 5 rate constants. Specifically, the fractions of total NBD-tagged lipid found in each pool were defined as follows: S = aqueous phase PS, $P_1$ = OMM outer leaflet PS, $P_2$ = OMM inner leaflet PS, and $P_3$ = OMM inner leaflet PE (Eq. 10). We took advantage of the environment sensitivity of NBD-PS fluorescence (~16-fold higher fluorescence of NBD-PS in a membrane environment compared with that of an NBD-PS monomer in solution (Fig. S12g)) to estimate the effective OMM adsorption and desorption rate constants ($k_0$ and $k_1$). Upon mixing NBD-PS with an excess of liposomes, fluorescence increased exponentially (Fig. S12h) with a rate set by the sum of both $k_0$ and $k_1$. The desorption rate constant $k_1$ was separately determined using BSA-back extraction with asymmetrically NBD-PS-labeled liposomes and excess BSA such that $k_1$ was rate-limiting, producing a single exponential time course (Fig. S12i)[77]. Taken together, these measurements provided estimates of $k_0$ ~ 1 s⁻¹ and $k_1$ ~ 0.05 s⁻¹. As NBD-PS was added to mitochondria above its critical micelle concentration (CMC) of 0.165 μM[78,79], aqueous NBD-PS will equilibrate between monomeric

[PS] and micellar [M] states. Only monomeric NBD-PS can be directly incorporated into lipid bilayers, and therefore the effective rate constant $k_0$ in our kinetic model represents a simplified composite of both the desorption of NBD-PS monomers from a micellar pool as well as their subsequent incorporation into membranes. Although the desorption rate constant $k_1$ may have been overestimated due to contributions by a micellar pool of NBD-PS driving the BSA-back extraction together with preloaded liposomes, this is unlikely as similar values of $k_1$ were measured when using NBD-PS concentrations both above and below the CMC, indicating that no residual micelles were present after NBD-PS incorporation into liposomes (Fig. S12j). For the lipid scramblase step, rate constants $k_2$ and $k_3$ were assumed to be equal as lipid scrambling is a passive bidirectional transport step. We also assumed that these rate constants are larger than 0.01 s⁻¹ in wild-type mitochondria based on previous studies[9]. To compare the $P_3$ pool in our 4-state model with kinetics of NBD-PE production in wild-type mitochondria, the differential equations (Eqs. 11–14) were solved numerically implementing the Bulirsch-Stoer method with Richardson extrapolation using custom-written routines in Igor Pro 8 (Wavemetrics). A broad range of values for parameters $k_2$, $k_3$, and $k_4$ were used for the numerical simulations and the best-fitting model produced $k_2 = k_3 = 0.027 ± 0.006$ s⁻¹ and $k_4 = 0.0018 ± 0.00005$ s⁻¹. As porin deletion would not be expected to alter $k_0$, $k_1$, or $k_4$, only $k_2$ and $k_3$ were adjusted in the kinetic model fits for the data obtained with *por1Δpor2Δ* mitochondria. The best-fitting model produced $k_2 = k_3 = 0.0025 ± 0.0004$ s⁻¹ in the mutant mitochondria, indicating that the lipid scrambling rate had dropped more than 10-fold in the absence of VDAC (Fig. 5d). Examples of the $P_3$ time course with wild-type and *por1Δpor2Δ* mutant rate constants are shown in Fig. 5c superimposed on the NBD-PE data points.

$$S \underset{k_1}{\overset{k_0}{\rightleftharpoons}} P_1 \underset{k_3}{\overset{k_2}{\rightleftharpoons}} P_2 \overset{k_4}{\rightarrow} P_3 \tag{10}$$

$$\frac{dS}{dt} = k_1 \cdot P_1 - k_0 \cdot S \tag{11}$$

$$\frac{dP_1}{dt} = k_0 \cdot S + k_3 \cdot P_2 - (k_1 + k_2) \cdot P_1 \tag{12}$$

$$\frac{dP_2}{dt} = k_2 \cdot P_1 - (k_3 + k_4) \cdot P_2 \tag{13}$$

$$\frac{dP_3}{dt} = k_4 \cdot P_2 \tag{14}$$

## Coarse-grained molecular dynamics simulations (CGMD)

**CGMD - General settings and systems preparation.** All MD simulations were performed using the simulation package Gromacs version 5.1.4[80] with plugin Plumed version 2.3[81]. We employed (i) coarse-grained Martini force-field version 3.0[82] and (ii) coarse-grained Martini force-field version 2.2[83–85] with ElNeDyn network[86]. For Martini 2 simulations of VDAC dimers, the Lennard-Jones interactions between protein beads were scaled down by 10% to prevent unrealistically strong protein-protein interactions[87]. We used a structure of human VDAC1 (hVDAC1) dimer solved by x-ray diffraction with a resolution of 2.7 Å and denoted as 6g6u in the RCSB Protein Data Bank (www.rcsb.org/structure/6g6u). A single monomer of VDAC1 was minimized in vacuum using atomistic Amber 99SB-ILDN force-field[88] with steepest descent algorithm until the maximum force on any atom was lower than 100 kJ mol⁻¹ nm⁻¹.

For Martini 3 simulations, the minimized atomistic protein structure was coarse-grained using Martinize2 script (https://github.

com/marrink-lab/vermouth-martinize) constructing an elastic network with a force constant of 1000 kJ mol$^{-1}$ nm$^{-2}$ to fix the tertiary structure of VDAC1. Dimeric structures were generated by placing the coarse-grained VDAC1 monomers at appropriate positions. No restraints maintaining the dimeric structure of VDAC1 were used unless explicitly stated otherwise. Each VDAC1 structure was embedded in the center of an xy-positioned membrane using the Insane script (https://github.com/Tsjerk/Insane). For simulations with monomeric VDAC1, the membrane was composed of roughly 660 molecules of 1-palmitoyl-2-oleoyl-sn-glycero-3-phosphocholine (POPC), while for the simulations of dimeric interfaces, the membrane consisted of roughly 920 molecules of POPC. Systems had an approximate size of roughly $15 \times 15 \times 12$ nm for simulations of monomeric VDAC1 and $18 \times 18 \times 11$ nm for dimer simulations and were solvated with roughly 16,000 and 20,000 water beads, respectively. KCl ions were added to each system at a concentration of 100 mM with an excess of ions to neutralize the system. K$^+$ ions were modeled as SQ5 particles with charge of +1 and mass of 39 a.u. Each constructed system was minimized using the steepest descent algorithm and a maximum force tolerance of 200 kJ mol$^{-1}$ nm$^{-1}$.

In Martini 2 simulations with ElNeDyn, the VDAC1 monomer and dimer structures were coarse-grained and embedded into the membrane using the CHARMM-GUI web interface[89]. The systems with monomeric and dimeric VDAC1 had similar sizes and the same molecular composition as the Martini 3 systems. In the case of Martini 2, the solvated K$^+$ ions were modeled as Qd-type particles with charge +1 (the same as Na$^+$ ions). The dimeric structures of VDAC1 were not fixed by any external potentials and all Martini 2 systems were minimized the same way as Martini 3 systems.

**CGMD - Equilibration and production simulations.** For both Martini 3 and Martini 2 simulations, the equilibration was performed in five stages with increasing simulation time-step (dt) and length (t): (I) dt = 2 fs, t = 0.5 ns; (II) dt = 5 fs, t = 1.25 ns; (III) dt = 10 fs, t = 1 ns; (IV) dt = 20 fs, t = 30 ns; (V) dt = 20 fs, t = 15 ns. In all, but the stage V, position restraints in the form of harmonic potential with a force constant of 1000 kJ mol$^{-1}$ nm$^{-2}$ were applied to all coordinates of protein backbone beads. For Martini 3 systems with VDAC1 dimers, water restraints were applied in stages I–III to keep water from entering the hydrophobic core of the membrane. For this purpose, inverted flat-bottom potential with a force constant (k) of 10 kJ mol$^{-1}$ nm$^{-2}$ and a reference distance (ref) of 2.2 nm from the membrane center of mass was applied to the z-coordinates of all water beads. Equilibration was performed in the NPT ensemble with the temperature being maintained at 300 K using stochastic velocity-rescaling thermostat[90] with a coupling constant of 1 ps. Water with ions, membrane, and protein(s) were coupled to three separate thermal baths. The pressure was kept at 1 bar using Berendsen barostat[91] with a coupling constant of 12 ps. The simulation box size was scaled semi-isotropically with a compressibility of $3 \times 10^{-4}$ bar$^{-1}$ in both xy-plane and z-direction.

To integrate Newton's equations of motion, we used the leap-frog algorithm. Non-bonded interactions were cut-off at 1.1 nm. Van der Waals potential was shifted to zero at the cut-off distance. The reaction field was used to treat electrostatic interactions with a relative dielectric constant of 15. For Martini 3 simulations, LINCS[92] parameters lincs-order and lincs-iter were set to 8 and 2, respectively, to avoid artificial temperature gradients[93]. Following equilibration, monomeric and several dimeric VDAC1 structures were simulated in Martini 3 for 10 μs using the same simulation settings as in stage V of the equilibration, except the Berendsen barostat was replaced with Parrinello-Rahman barostat[94]. The same simulation settings were also used for other simulations described in the section CGMD - Free energy calculations. Three independent replica simulations were performed for each of the systems with monomeric or dimeric VDAC1. Apart from unrestrained simulations, we simulated the fastest scrambling dimer,

dimer-1, restrained to its initial structure using harmonic potential (k = 1000 kJ mol$^{-1}$ nm$^{-2}$) in the x- and y-coordinates of backbone beads of the glutamate 73 and serine 193 of both VDAC1 protomers.

**CGMD - Analysis.** For each of the three independent MD simulations, we calculated the percentage of scrambled lipids in time, analyzing a simulation snapshot every 10 ns. Lipid was identified as "scrambled" if it was positioned in a different membrane leaflet than at the start of the simulation. Lipids were assigned to the leaflet by the position of their phosphate bead with respect to the membrane center. For further analysis, all three independent replicas were concatenated and centered on protein beads using its root-mean-square deviation. We constructed a water defect map describing membrane disruption around the protein structure. Water defect was defined as the average number of water beads located closer to the membrane center on the z axis than 1 nm. The average number was calculated using xy-mesh with 0.1 nm bin every 100 ps. The same mesh was employed to calculate the average membrane thickness using z-positions of lipid phosphates. The average membrane thickness was defined as the difference between the position of the upper- and lower-leaflet phosphates. The bins with less than 100 samples were excluded from the analysis. The code used for the analysis of lipid scrambling and membrane disruption is available from [https://github.com/Ladme/scramblyzer] and [https://github.com/Ladme/memdian], respectively. The position of the beta-strands was approximated by the average position of the backbone bead of the central residue in the xy-plane.

**CGMD - Free energy calculations.** We calculated the free energy of lipid flip-flop in several systems: (a) in protein-free POPC membrane (Martini 3 & Martini 2), (b) in system with VDAC1 monomer (Martini 3 & Martini 2), (c) in system with VDAC1 dimer-1 (Martini 3), (d) in system with VDAC1 monomer while turning off the attractive Lennard-Jones (LJ) interactions between the scrambling lipid and VDAC (Martini 3), and (e) in system with VDAC1 dimer-1 while turning off the attractive Lennard-Jones interactions between the scrambling lipid and VDAC (Martini 3).

To enhance the sampling of the lipid flip-flop, we employed the umbrella sampling method[95]. For free energy calculations in the presence of VDAC1 monomer, Hamiltonian replica exchange was further applied[96]. Three one-dimensional collective variables (CVs) were used to capture the lipid flip-flop in different systems. For simulations without the presence of VDAC1, the oriented z-distance between a chosen lipid phosphate bead and the local membrane center of mass was used as a CV. The local membrane center of mass was calculated from the positions of lipid beads localized inside a cylinder with a radius of 2.0 nm and its principal axis going through the selected phosphate bead. For simulations of systems with VDAC1 monomer, the CV was defined as the oriented z-distance between a chosen lipid phosphate bead and backbone bead of glutamate 73. In systems with VDAC1 dimer-1, the CV was defined as the oriented z-distance between a chosen lipid phosphate bead and the center of mass of the dimer-1 which was positioned close to the center of the membrane on the z-axis. During the pulling and umbrella sampling simulations, the dimer-1 was restrained in the same way as described in the section CGMD - Equilibration and production simulations.

Initial configurations for umbrella sampling were generated by pulling the chosen lipid phosphate bead through the membrane for 1 μs with a pulling rate of 4.2 nm μs$^{-1}$ and initial reference distance of 2.1 (for systems without VDAC1) or ±2.0 (for systems with VDAC1) nm using a harmonic potential. In all systems, one pulling simulation was run for each direction of pulling (from the upper to the lower membrane leaflet and in the opposite direction). The exception was Martini 3 systems with VDAC1 monomer, where two pulling simulations in each direction were performed to generate a larger ensemble of initial configurations for umbrella sampling with Hamiltonian replica

 

exchange. In monomeric VDAC1 systems, the pulled phosphate was restrained by an xy-plane flat-bottom potential (k = 500 kJ mol$^{-1}$ nm$^{-2}$, ref = 1.5 nm) to stay close the backbone bead of the glutamate 73 residue. In dimeric VDAC1 systems, the phosphate was restrained by an xy-plane flat-bottom potential (k = 500 kJ mol$^{-1}$ nm$^{-2}$, ref = 2.5 nm) to stay close to the dimeric interface.

For systems with VDAC1, two independent sets of umbrella sampling windows were prepared to observe any possible hysteresis of the flip-flop process. For Martini 2 simulations, the initial configurations for each set of umbrella sampling windows were obtained from the pulling simulation performed in the specific direction and the calculation was enhanced by applying Hamiltonian replica exchange in these windows. In Martini 3 simulations with VDAC1 monomer, the efficiency of the Hamiltonian replica exchange was further enhanced by using configurations from two pulling simulations performed in the same direction. For systems without VDAC1, only one pulling direction was used.

We used between 15 (Martini 3 simulations of dimer-1) and 72 windows (Martini 2 simulations of monomeric VDAC) distributed along the range of the CV for each pulling direction. See Tables 1–5 for the complete list of umbrella sampling windows that were used for the calculations. After a short 30 ns equilibration, each window was sampled for a specific time which differed based on the system sampled and force-field used. The simulated times necessary to obtain converged free energy profiles (Fig. S13) were (a) 1 μs for protein-free membrane, (b) 4 or 8 μs for simulations with VDAC1 monomer in Martini 3 and Martini 2, respectively, (c) 6 μs for simulations with VDAC1 dimer, (d) 2 μs for simulations with VDAC1 monomer and attractive LJ interactions turned off, and (e) 4 μs for simulations with VDAC1 dimer and attractive LJ interactions turned off. In simulations in which Hamiltonian replica exchange was applied, the exchange was attempted every 10,000 integration steps (200 ps). Free energy profiles were obtained from the umbrella sampling windows using the weighted histogram analysis method[97,98] as implemented in the Gromacs tool g_wham[99].

The initial structure for the dimer-1-mutant was generated by taking the structure of the wild-type VDAC1 monomer and replacing residues T77, S43, T33, S35, Y247, and Q249 with valines using MODELLER version 9.11[100]. The mutated monomeric structure was then minimized in vacuum, coarse-grained and a dimer-1 version was prepared in the same way as described above. Equilibration and production simulations and analysis were performed in the same way as for wild-type dimer-1 (described above).

**Reporting summary**
Further information on research design is available in the Nature Portfolio Reporting Summary linked to this article.

## Data availability
The data that support this study are available from the corresponding authors upon request. Molecular dynamics simulation trajectories input files can be accessed at Zenodo [https://doi.org/10.5281/zenodo.8358517]. Programs for analyzing the simulations can be found in a separate Zenodo archive [https://doi.org/10.5281/zenodo.8360034]. A Source Data file is available with this manuscript. Source data are provided with this paper.

## Code availability
The code used for the analysis of lipid scrambling and membrane disruption is available in public github repositories [https://github.com/Ladme/scramblyzer] and [https://github.com/Ladme/memdian].

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

## Acknowledgements

We thank Steve Claypool (Johns Hopkins University) for antibodies, Non Miyata (Kyushu University) for yeast strains, Matthew Johnson and Denisse Leyton (Australian National University) for a sample of Pet464 protein and the Pet464 expression construct, Kathryn Diederichs and Susan Buchanan (NIDDK) for the Por1 expression construct, Toon de Kroon (Utrecht University) for the mitochondria isolation protocol, and Bob Dylan for too much of nothing. This work was supported by National Institutes of Health grants NS119779 (AKM) and NS116747 (J.S.D.), Boehringer Ingelheim Fonds Ph.D. Fellowship (H.J.), Deutsche Forschungsgemeinschaft grants SFB944-P14 and HO 3539/2-1 (J.C.M.H.), Czech Science Foundation grant GA20-20152S (R.V.), European Research Council (ERC) under the European Union's Horizon 2020 research and innovation program (grant agreement No 101001470) (R.V.), and National Institute of virology and bacteriology (Programme EXCELES, ID Project No. LX22NPO5103) - Funded by the European Union - Next Generation EU (R.V.). Computational resources were provided by the CESNET LM2015042 and the CERIT Scientific Cloud LM2015085 under the program Projects of Large Research, Development, and Innovations Infrastructures. Additional computational resources were obtained from IT4 Innovations National Supercomputing Center --

LM2015070 project supported by MEYS CR from the Large Infrastructures for Research, Experimental Development, and Innovations.

## Author contributions

Conceptualization: H.J., J.C.M.H., J.S.D., R.V., A.K.M. Methodology: H.J., L.B., G.I.D., J.S.D., R.V., A.K.M. Investigation: H.J., L.B., G.I.D. Visualization: H.J., L.B., R.V., A.K.M. Funding acquisition: H.J., J.S.D., J.C.M.H., R.V., A.K.M. Project administration: R.V., A.K.M. Supervision: R.V., A.K.M. Writing – original draft: H.J., L.B., A.K.M. Writing – review & editing: H.J., L.B., G.I.D., J.S.D., J.C.M.H., R.V., A.K.M.

## Competing interests

The authors declare no competing interests.
