## [Peer Review File · Nature Communications]

Phospholipids are imported into mitochondria by VDAC, a dimeric beta barrel scramblaseReviewers' Comments:

Reviewer #1:

Remarks to the Author:

This manuscript very convincingly reports the role of the voltage dependent anion channel (VDAC) as a lipid scramblase in the outer mitochondrial membrane (OMM), for which no scramblase was yet identified - interestingly, this is the first report of a β -barrel protein being able to scramble lipids. VDAC is the most abundant protein of the OMM, with various roles from metabolites permeation to mitochondrial regulation (calcium homeostasis, apoptosis, neurodegeneration). Lipid scrambling comes as another function for VDAC, and is well in agreement with previous studies, cited by the authors, on the ability of VDAC to influence the membrane thickness.

The authors demonstrated clearly the utmost importance of oligomerization of both VDAC1 and VDAC2 for lipid scrambling, using a combination of biochemistry and molecular dynamics techniques. VDAC oligomerization is currently a very discussed topic in the VDAC field, as it seems that different oligomeric forms of VDAC are necessary to perform different function in the OMM.

The results show lipid scrambling result as a combination of membrane thinning at the dimer interface, and a network of polar residues facing the membrane allowing the polar head group to change leaflet.

Additional experiments on purified yeast mitochondria allowed to validate their finding in a more physiological environment.

The results presented in this well written manuscript are very important both for the lipid transport and the VDAC fields. To my opinion, this is by far one of the best papers on VDAC in those last 5 years, and it definitely fits in a journal of the level of Nature Communication.

The authors investigated isoforms 1 and 2 of VDAC, but not VDAC3 (that is present in humans but not in yeast). It would be interesting to explore VDAC3's ability to scramble lipids. However, this isoform is a difficult protein to produce and refold under an active form, and could be the object of a follow up study.

Here are a few points on the manuscript that should be addressed:

Page 6, MD simulation.

"We chose two symmetric dimers, with interfaces mediated by strands b1,2,18,19 (dimer-1) (Fig. 4A,B) as previously reported 29,37,38, and b14-17 (dimer-3), a novel configuration in which the E73 residue is positioned distal to the interface in each protomer."

Where does Dimer 3 come from ? Was it previously published?

"Unless restrained, dimer-1 re-oriented during the simulation to dimer-1*, with a symmetric interface mediated by strands b2-4 (Fig. S8B), at which point the rate fell \sim 4-fold. "

This dimer is highly similar to the one described in the study of Bergdoll et al. (PNAS 2017), the authors should acknowledge it and reference this paper here.

Please specify the VDAC1-5V mutations once in this paragraph, as different amino acids are specified in the methods:

P. 10: VDAC1 and VDAC2 isoforms, the point mutant VDAC2 E84L, or the pentamutant VDAC1-5V (T60V, Y62V, E73V, T83V, S101V).

P. 19: The initial structure for the VDAC1-5V mutant was generated by taking the structure of wildtype VDAC1 monomer and replacing residues T77, S43, T33, S35, Y247, and Q249 with valines.

P. 7, the first paragraph of this page is a little difficult to understand. Fig S.10 could benefit from a legend on the figure to make it easier to read, and could be explained a little more in the text.

(There is no C on Fig S10. Page 44).

Reviewer #2:

Remarks to the Author:

Jahn et al describe a new kind of lipid scramblase built from beta-barrel transmembrane proteins and functioning in the outer mitochondrial membrane. This is a beautiful manuscript. The biochemical standards needed to define a protein as a "scramblase" have largely been developed within the Menon lab, and so it is no surprise to see biochemical activity of VDAC2 established in multiple different assays. Beyond that however, the group also reconstitutes several scramblase-dead or deficient proteins which provide tremendous confidence in the approach; they reconstitute the yeast VDAC homolog (Por1) and demonstrate activity consistent with a biologically-relevant activity; they characterize the path of lipid scrambling and introduce mutants that specifically disrupt that path (by altering dimerization) establishing the mechanism; and most important of all, they introduce gain-of-function mutations into a minimally active VDAC1, forcing it into a dimer and showing the emergence of lipid scrambling. This is a real tour-de-force.

The groups use molecular dynamics to describe how the dimer interface thins the membrane and accelerates lipid movement. Their biochemistry mirrors the outputs from the MD simulations strongly implying that the model is relevant.

Finally, they look for evidence of VDAC-dependent scrambling in intact mitochondria. The assay is indirect, they follow the production of lipids within the mitochondria that depend upon transport from outside. The total change in lipid synthesis is fairly subtle, but the authors make a compelling case that scrambling is unlikely to be the rate-limiting event and account for each of the key steps in lipid movement/synthesis in a simple kinetic model with constants derived from experimental data in simple systems. There are, necessarily, some assumptions built in such as the idea that the partition into/out of liposomes sufficiently mimics the same event in isolated mitochondria and that these events are independent of the presence of VDAC subunits, but overall the resulting model is an elegant demonstration of the role of these proteins in mitochondrial lipid synthesis.

I have no major critiques. Two very minor thoughts – could the authors say more about the limitations of their kinetic model? As other scramblases are bound to be discovered on the mitochondrial membrane (they reveal one such example in their discussion), it seems likely there may be a tremendous untapped scrambling potential and thus how/why loss of Por1 leads to a 90% reduction in scrambling will be likely be contested in the future. Second, the section "The VDAC pore is not relevant for phospholipid scrambling" features a discussion surrounding how if the pore was important VDAC1 would be as good as VDAC2 at scrambling. This make sense to me. But then they introduce a new scrambling experiment which relies upon lipid deposition more than just extraction and show, again, that VDAC1 is a weak scrambler. I am not sure this experiment is any more compelling than the earlier examples in the paper and do not see how it is uniquely suited to the question of whether the pore is important. I might suggest that rather than a full sub-section on this topic, it is simply discussed at the culmination of all of the scrambling data (including the exogenous NBD-PC assay).

Reviewer #3:

Remarks to the Author:

1. The authors have produced VDAC1 and VDAC2, mitochondrial membrane proteins with a β -barrel structure, in *E. coli*. These proteins were initially recovered as inclusion bodies and subsequently denatured and renatured. In the Results section, the authors claim that the protein was "functionally" folded (Line 3). While the circular dichroism (CD) spectra support the correctness of the secondary structure, it does not provide information about the correctness or functionality of the tertiary

structure. Furthermore, the size exclusion chromatography (SEC) profile in Figure S2 suggests that the purified protein is not monomeric (with an estimated molecular weight of 30,746 for VDAC), as it elutes at the void volume of the Superdex 200 column. To confirm the monomeric state, running the sample on a Superose 6 column with molecular weight standards or analysing it using Blue-native PAGE is recommended.

2. The authors have reconstituted VDAC1/VDAC2 into artificial lipid vesicles with a 90% PC and 10% PG composition. However, it is important to note that the lipid composition of mitochondrial outer membranes significantly differs from the composition used in this reconstitution. As the mitochondrial outer membrane primarily consists of PC, PE, PI, and other lipids, it is crucial to reconstitute mitochondrial proteins in membranes with lipid compositions that better resemble those found in mitochondria. This step will enhance the relevance of the experimental setup to the natural environment of VDAC1/VDAC2.

3. The authors propose that VDAC forms a dimer, with a specific dimeric form functioning as a scramblase. To validate this claim, it is necessary to provide evidence of native (non-cross-linked) protein dimerization. This can be achieved through gel filtration or Blue-native PAGE, demonstrating the presence of dimeric forms without chemical cross-linkers. Also, it is recommended to be biochemically to confirm the cross-linker's connection to specific residues of VDAC to confirm the proposed dimeric structure.

4. Regarding Figure S12, it is advisable to adjust the y-axis to start from 0 instead of 10 for clarity and accuracy in data representation. Additionally, the transformation of the NBD-PS data from Fig. 5C into Fig. 5D should be explained in more detail to enhance understanding. Given that the authors assert that VDAC's scramblase activity does not require ATP or energy (Page 2, the second paragraph), it would be valuable to observe the scrambling activity of OMM's VDAC at 4°C, where the effect of NBD-PS decarboxylation, typically requiring energy, can be minimized. Please note that TMEM16F at the plasma membranes functions at 4°C (PMID: 28559311).

5. The authors assert that Por1 and Por2 (VDAC1 and VDAC2 in yeast) are the primary scramblases in mitochondrial outer membranes. In light of this, it is essential to investigate the broader impact of Por1 and Por2 on the composition of other phospholipids in the membrane. A group previously reported reduced Cardiolipin content in the mitochondria of Por1-Por2 mutant yeast (PMID: 30237174). To gain a more comprehensive understanding, it would be beneficial to assess how the composition of other phospholipids is affected by the presence or absence of Por1 and Por2. This information would contribute to a more comprehensive analysis of lipid trafficking in the mitochondrial outer membrane.

POINT-BY-POINT RESPONSE TO REVIEWERS' COMMENTS

REVIEWER #1

1. Page 6, MD simulation. "We chose two symmetric dimers, with interfaces mediated by strands b1,2,18,19 (dimer-1) (Fig. 4A,B) as previously reported 29,37,38, and b14-17 (dimer-3), a novel configuration in which the E73 residue is positioned distal to the interface in each protomer." Where does Dimer 3 come from ? Was it previously published?

→ Dimer 3 has not been previously described (we used it to establish that a random interface is not sufficient to enable lipid scrambling) and to make this clear, we now refer to dimer-3 as a "novel synthetic configuration" on page 6.

2. "Unless restrained, dimer-1 re-oriented during the simulation to dimer-1*, with a symmetric interface mediated by strands b2-4 (Fig. S8B), at which point the rate fell ~4-fold. "

This dimer is highly similar to the one described in the study of Bergdoll et al. (PNAS 2017), the authors should acknowledge it and reference this paper here.

→ We thank the reviewer for raising this point and have included it in the manuscript with a citation to Bergdoll et al. PNAS 2018 on page 6 as follows: "... dimer-1*, resembling the dimer previously reported by Bergdoll et al. ³¹, with a symmetric interface mediated by strands β 2-4."

3. Please specify the VDAC1-5V mutations once in this paragraph, as different amino acids are specified in the methods: P. 10: VDAC1 and VDAC2 isoforms, the point mutant VDAC2 E84L, or the pentamutant VDAC1-5V (T60V, Y62V, E73V, T83V, S101V). P. 19: The initial structure for the VDAC1-5V mutant was generated by taking the structure of wildtype VDAC1 monomer and replacing residues T77, S43, T33, S35, Y247, and Q249 with valines.

→ We thank the reviewer for pointing out this error in our description. We have cleared up the inconsistency. VDAC1-5V is a penta-mutant (T60V, Y62V, E73V, T83V, S101V) which was tested experimentally (Fig. S9). The mutant referred to on page 19 is a hexa-mutant (T77V, S43V, T33V, S35V, Y247V, Q249V) which was used in the MD simulation described in Fig. S8 - we now consistently refer to the hexa-mutant as 'dimer-1-mutant'.

4. P. 7, the first paragraph of this page is a little difficult to understand. Fig S.10 could benefit from a legend on the figure to make it easier to read and could be explained a little more in the text. (There is no C on Fig S10. Page 44).

→ We have re-written the paragraph to make it clearer (bottom of page 6) and added labels to the panels in Fig. S10.

REVIEWER #2

1. Could the authors say more about the limitations of their kinetic model? As other scramblases are bound to be discovered on the mitochondrial membrane (they reveal one such example in their discussion), it seems likely there may be a tremendous untapped scrambling potential and thus how/why loss of Por1 leads to a 90% reduction in scrambling will be likely be contested in the future.

→ The reviewer raises a very interesting point. Although our model is highly simplified, we chose conservative values for the rate constants that likely underestimate the true scrambling rate. Thus, it is reasonable to conclude from these efforts and experiments that 90% is an underestimate of the true contribution by VDAC. In the Discussion section, we suggested the presence of other scramblases, including yeast Ugo1 (mammalian MTCH2) and yeast Mcp1 (no mammalian counterpart), that might account for the activity remaining after removal of VDACs (we now cite a new preprint reporting the scramblase activity of purified MTCH2). Approximate protein abundance as reported on the Saccharomyces Genome Database is Por1 (20,000 copies per cell), Por2 (2000), Ugo1/MTCH2 (1500), Mcp1 (no data, possibly very low abundance in wild-type cells). The contributions of Por2, Ugo1/MTCH2 and Mcp1 are thus likely to be individually much smaller than that of Por1, if the unitary rate for all proteins is the same. Indeed, our data indicate that deletion of Por1 and Por2 together does not significantly increase the magnitude of the scrambling defect seen with Por1 deletion alone.

To incorporate these points, we have added/revise the text on pages 8 (Results) and 9 (Discussion) as follows:

Page 8 - 'Although our kinetic model is highly simplified, we chose conservative values for the rate constants that likely underestimate the true scrambling rate in wild-type mitochondria. The loss of >90% of the lipid import capability in mitochondria lacking VDACs therefore indicates that VDAC proteins provide the majority of scramblase activity associated with the OMM.'

Page 9 - 'Further work will be needed to quantify the relative individual contributions of MTCH2 and the other scramblases mentioned above to the overall scramblase activity in the OMM.'

2. The section "The VDAC pore is not relevant for phospholipid scrambling" features a discussion surrounding how if the pore was important VDAC1 would be as good as VDAC2 at scrambling. This makes sense to me. But then they introduce a new scrambling experiment which relies upon lipid deposition more than just extraction and show, again, that VDAC1 is a weak scrambler. I am not sure this experiment is any more compelling than the earlier examples in the paper and do not see how it is uniquely suited to the question of whether the pore is important. I might suggest that rather than a full sub-section on this topic, it is simply discussed at the culmination of all of the scrambling data (including the exogenous NBD-PC assay).

→ We thank the reviewer for this suggestion, which helps to streamline the presentation. We have eliminated the stand-alone section on the non-relevance of the VDAC pore and used some of the text to introduce the MD simulations section on page 5.

REVIEWER #3

1. The authors have produced VDAC1 and VDAC2, mitochondrial membrane proteins with a β -barrel structure, in *E. coli*. These proteins were initially recovered as inclusion bodies and subsequently denatured and renatured. In the Results section, the authors claim that the protein was "functionally" folded (Line 3).

While the circular dichroism (CD) spectra support the correctness of the secondary structure, it does not provide information about the correctness or functionality of the tertiary structure.

→ We believe that our data and literature precedents clearly indicate that the VDAC1, VDAC2 and Por1 proteins described in the paper are functional β -barrel channels. We purified VDAC1 and VDAC2 from inclusion bodies following the protocol of Dadsena et al. (Nature Commun 2019, PMID 31015432) which is similar to standard protocols that have been extensively reported in the literature to prepare VDAC for structural and other analyses (please see below). We verified the quality of our purified proteins by CD spectroscopy and demonstrated that they form channels after reconstitution into liposomes. These key data confirm that VDAC1 and VDAC2, prepared as we describe, are functional β -barrel channels. We also

provide data on yeast Por1 (Fig. S11) which was purified after expression in yeast under 'native' conditions. This protein was extracted from yeast directly into LDAO, purified (no denaturation or renaturation steps were used) and reconstituted, demonstrating both channel and scramblase activity. These results corroborate our findings with bacterially expressed human VDAC proteins.

As noted above, our protocol builds on abundant precedents in the literature. Most investigations, including ours, make use of LDAO as the detergent of choice for refolding. For example, Engelhardt et al. (J Membr Biol 2007, PMID 17828567) used a suite of zwitterionic and nonionic detergents to refold VDAC1 and VDAC2 after denaturation from inclusion bodies, concluding - by a variety of analytical techniques - that LDAO is the best. An NMR study of VDAC2 (Yu et al. (BBA 2012, PMID 22119777) prepared the protein exactly as we describe to obtain structural information in both LDAO micelles and after reconstitution into lipid nanodiscs, concluding that the protein is properly folded in both detergent and nanodisc systems. A similar procedure was used to obtain an NMR structure of human VDAC1 (Bayrhuber et al. PNAS 2008, PMID 18832158), a crystal structure of zebra fish VDAC2 (Schredelseker et al. JBC 2014, PMID 24627492), and electrophysiological data after reconstituting mouse VDAC1 in planar bilayers (Queralt-Martin et al. BBA 2019, PMID 30412693). These are just a few of the many publications that provide extensive documentation of the structure and functionality of VDAC preparations, generated as per the general protocol that we describe.

Furthermore, the size exclusion chromatography (SEC) profile in Figure S2 suggests that the purified protein is not monomeric (with an estimated molecular weight of 30,746 for VDAC), as it elutes at the void volume of the Superdex 200 column. To confirm the monomeric state, running the sample on a Superose 6 column with molecular weight standards or analysing it using Blue-native PAGE is recommended.

→ We thank the reviewer for raising this point as it prompted us to add clarifications to the manuscript. Briefly, VDAC proteins chromatograph within the Superdex 200 column, not at the void volume. Using detergent conditions that mimic our experimental protocols, we show that VDAC1 can be recovered, in part, as a monomer. This information is now included in the revised manuscript in the form of SEC data in new Fig. S7A and text changes on page 5. We note that our paper describes the channel and scramblase activity of VDAC proteins in a membrane context, and that SEC analyses - while generally informative - cannot report on the quaternary structure of VDAC in the membrane. We provide a detailed response to the reviewer's comment below.

Size exclusion was used as a polishing step in our VDAC purification protocol. VDAC1 runs within the resolving portion of the Superdex 200 column, coinciding with the migration of the apoferritin (440 kDa) standard. The size exclusion chromatogram for VDAC1 shown in Fig. S2A, is reproduced along with a series of molecular weight markers and blue dextran (to indicate void volume) in revised Fig. S7A.

The SEC profile in Fig. S2A was run using 0.05% LDAO, a condition in which VDAC is multimeric. When LDAO concentration is increased, the multimers dissociate into smaller oligomers and monomers. We incubated VDAC1 in 1% LDAO and re-analyzed the protein by SEC (run in 1% LDAO) - Fig. S7A shows that under these conditions, VDAC1 chromatographs as two peaks corresponding to a monomer (VDAC1 together with an LDAO micelle would be expected to run at approximately 50 kDa, between the 29 kDa and 66 kDa standards) and also a slightly larger species which could be a dimer or trimer. In an intermediate LDAO concentration (0.2%), the proportion of these two peaks shifts such that the dimer/trimer peak is the predominant one (not shown). An example of similar data can be found in Zalk et al. (Biochem J 2005, PMID 15456403) where samples were prepared in the presence of high amounts of detergent to dissociate VDAC oligomers into monomers, followed by detergent withdrawal to promote re-association.

The SEC data in Fig. S7A obtained with low and high LDAO concentrations are supported by the EGS crosslinking results in Fig. S7B which show that VDAC multimers can be captured at a low LDAO

concentration, but that monomers prevail at high LDAO concentrations. Finally, our results are corroborated by the reconstitution experiment shown in Fig. 2C. Here VDAC proteins are added to destabilized vesicles in high LDAO, but then integrate functionally (determined by channel assay) into vesicles as multimers as detergent is removed (Fig. 2C).

We believe that the state of the protein under different conditions is now well-documented in the paper - including new Fig. S7A - and supported by published precedents.

2. The authors have reconstituted VDAC1/VDAC2 into artificial lipid vesicles with a 90% PC and 10% PG composition. However, it is important to note that the lipid composition of mitochondrial outer membranes significantly differs from the composition used in this reconstitution. As the mitochondrial outer membrane primarily consists of PC, PE, PI, and other lipids, it is crucial to reconstitute mitochondrial proteins in membranes with lipid compositions that better resemble those found in mitochondria. This step will enhance the relevance of the experimental setup to the natural environment of VDAC1/VDAC2.

→ POPC/POPG 9:1 is a standard lipid mixture used in many reconstitution studies. Although we could instead use a lipid mixture corresponding to that reported for the mitochondrial outer membrane as suggested by the reviewer, this would not be sufficient to reach specific conclusions about the role of particular lipids in VDAC's scramblase function. This is because vesicles prepared from several lipid components are compositionally heterogeneous (PMID 21688773, 22239728, 37308687) with individual vesicles having quite different compositions; this makes bulk, ensemble measurements, such as our channel and scramblase activity assays, difficult to interpret. Also problematic, is that it is not possible to mimic the transbilayer lipid asymmetry of the outer membrane, even if a compositionally exact, uniform population of vesicles could be generated. Rather than pursue this approach, we instead assayed scramblase activity directly in the natural lipid environment of mitochondria (Figs. 5 and S12).

3. The authors propose that VDAC forms a dimer, with a specific dimeric form functioning as a scramblase. To validate this claim, it is necessary to provide evidence of native (non-cross-linked) protein dimerization. This can be achieved through gel filtration or Blue-native PAGE, demonstrating the presence of dimeric forms without chemical cross-linkers.

→ As our proposal concerns the dimeric state of VDAC in liposomes, and not detergent, neither gel filtration, BN-PAGE or co-purification are ultimately useful to provide relevant evidence of dimerization, whereas crosslinking experiments are commonly used for this purpose (for example, Kim et al. Science 2021, PMID 31857488). Please see below for a detailed response.

Both VDAC1 and VDAC2 form native multimers as seen in the SEC profiles shown in Fig. S2 where the proteins migrate approximately with the 440 kDa standard when analyzed in 0.05% LDAO, and a VDAC dimer or trimer is observed when the LDAO concentration is increased (Fig. S7A, see above for discussion of this point). In support of the points raised above, endogenous Por1 co-purifies with expressed, Twin-Strep-tagged Por1 when isolated from yeast, indicating association of natively extracted proteins in LDAO (Fig. S11A).

Both VDAC1 and VDAC2 reconstitute as multimers (Fig. 2C), based on the functionalization of vesicles with channels. Analysis of the reconstituted samples by crosslinking reveals that VDAC2 remains significantly dimeric/multimeric after reconstitution (Fig. 2G), whereas VDAC1 largely dissociates and cannot be efficiently captured in higher order forms by crosslinker, although a small dimeric population is evident (Fig. 2H). In these experiments (Fig. 2G,H) the EGS crosslinker is used to detect pre-existing dimers/multimers in liposomes and not to generate them.

Our claim that specific VDAC1 dimers have scramblase activity is validated by our experimental data on VDAC1-5V which forms dimers/oligomers spontaneously, without the necessity of crosslinking. These dimers - detected in liposomes by using a crosslinker (Fig. S9A) - have poor scramblase activity (Fig. S9C,D)

clearly indicating that not all dimers are functional. Additionally, our MD simulation data show that several dimers, including dimer-1*, dimer-1-mutant and dimer-3 are all relatively inactive as scramblases compared with dimer-1 (Fig. 4C,D, Fig. S8C-E).

Also, it is recommended to be biochemically to confirm the cross-linker's connection to specific residues of VDAC to confirm the proposed dimeric structure.

→ Evidence of native, non-crosslinked VDAC dimers is available in numerous reports. Structural studies of VDAC proteins reveal parallel dimers with an interface mediated by beta-strands 17-19, 1-3 (for example, VDAC1 - Bayrhuber et al. PNAS 2008, PMID 18832158, and VDAC2 - Schredelseker et al. JBC 2014, PMID 24627492). As these dimeric proteins were prepared using the protocol that we used here (see response to point #1), we anticipate that we are working with the same dimeric form with an interface mediated by beta-strands 17-19, 1-3. Indeed, this is the interface that we used to generate the dimer-1 construct for our MD simulations, showing that is active as a scramblase whereas other dimers are not. The reviewer recommends mapping the residues implicated in chemical cross linking - this is far from trivial and something that is the objective of future work. We note that this was not a request from the reviewer, but rather a recommendation. Nevertheless, in Fig. S5, we suggest which pairs of lysine residues are likely to participate in the crosslinks.

4. Regarding Figure S12, it is advisable to adjust the y-axis to start from 0 instead of 10 for clarity and accuracy in data representation. Additionally, the transformation of the NBD-PS data from Fig. 5C into Fig. 5D should be explained in more detail to enhance understanding. Given that the authors assert that VDAC's scramblase activity does not require ATP or energy (Page 2, the second paragraph), it would be valuable to observe the scrambling activity of OMM's VDAC at 4°C, where the effect of NBD-PS decarboxylation, typically requiring energy, can be minimized. Please note that TMEM16F at the plasma membranes functions at 4°C (PMID: 28559311).

→ We have modified Fig. S12F so that the y-axis starts at 0 instead of 10. We have also clarified the methods section 'Analysis of the kinetics of transport-coupled PS decarboxylation by yeast mitochondria' by adding explanatory text and the reaction scheme and equations that define the 4-pool kinetic model which we used to fit the raw data in Fig. 5C to obtain the line fits shown in Fig. 5C and scrambling rates shown in Fig. 5D.

We do not understand the reviewer's suggestion to carry out the transport-coupled NBD-PS decarboxylation assay at 4°C. As Psd1-catalyzed decarboxylation provides the read-out of scrambling in this assay, it is not clear what will be learned about scrambling if the Psd1 reaction, which is rate-limiting in wild-type mitochondria, is slowed even further by lowering the temperature. We note that it would be analytically advantageous to increase the rate of the Psd1 reaction, not slow it down.

5. The authors assert that Por1 and Por2 (VDAC1 and VDAC2 in yeast) are the primary scramblases in mitochondrial outer membranes. In light of this, it is essential to investigate the broader impact of Por1 and Por2 on the composition of other phospholipids in the membrane. A group previously reported reduced Cardiolipin content in the mitochondria of Por1-Por2 mutant yeast (PMID: 30237174). To gain a more comprehensive understanding, it would be beneficial to assess how the composition of other phospholipids is affected by the presence or absence of Por1 and Por2. This information would contribute to a more comprehensive analysis of lipid trafficking in the mitochondrial outer membrane.

→ We agree with the reviewer and have cited several papers, including the one mentioned, in the second paragraph of the Discussion section on page 9: PMID 30237174, 34572064, 37636069. These papers attest to the dysregulation of overall lipid composition in cells with VDAC/porin deficiency. Of special note is the decrease in the levels of cardiolipin, which is specific to mitochondria.

Reviewers' Comments:

Reviewer #1:

Remarks to the Author:

All my comments have been addressed by the authors. The revisions improved the paper. In my opinion, it is ready for publication.

Reviewer #2:

Remarks to the Author:

I am fully satisfied by the author's updated manuscript. Very nice piece of scholarship!